Corrected: Publisher correction

# A $Co_3O_4$-CDots-$C_3N_4$ three component electrocatalyst design concept for efficient and tunable $CO_2$ reduction to syngas

Sijie Guo[1], Siqi Zhao[1], Xiuqin Wu[1], Hao Li[1], Yunjie Zhou[1], Cheng Zhu[1], Nianjun Yang[2],

Xin Jiang[2], Jin Gao[1], Liang Bai[1], Yang Liu[1], Yeshayahu Lifshitz[1,3], Shuit-Tong Lee[1] & Zhenhui Kang[1]

Syngas, a CO and $H_2$ mixture mostly generated from non-renewable fossil fuels, is an essential feedstock for production of liquid fuels. Electrochemical reduction of $CO_2$ and $H^+/H_2O$ is an alternative renewable route to produce syngas. Here we introduce the concept of coupling a hydrogen evolution reaction (HER) catalyst with a CDots/$C_3N_4$ composite (a $CO_2$ reduction catalyst) to achieve a cheap, stable, selective and efficient route for tunable syngas production. $Co_3O_4$, $MoS_2$, Au and Pt serve as the HER component. The $Co_3O_4$-CDots-$C_3N_4$ electrocatalyst is found to be the most efficient among the combinations studied. The $H_2$/CO ratio of the produced syngas is tunable from 0.07:1 to 4:1 by controlling the potential. This catalyst is highly stable for syngas generation (over 100 h) with no other products besides CO and $H_2$. Insight into the mechanisms balancing between $CO_2$ reduction and $H_2$ evolution when applying the HER-CDots-$C_3N_4$ catalyst concept is provided.

[1] Jiangsu Key Laboratory for Carbon-Based Functional Materials & Devices, Institute of Functional Nano & Soft Materials (FUNSOM), Soochow University, 199 Ren'ai Road, Suzhou 215123 Jiangsu, China. [2] Institute of Materials Engineering, University of Siegen, 57076 Siegen, Germany. [3] Department of Materials Science and Engineering, Technion, Israel Institute of Technology, Haifa 3200003, Israel. Correspondence and requests for materials should be addressed to X.J. (email: xin.jiang@uni-siegen.de) or to Y. Liu (email: yangl@suda.edu.cn) or to Y. Lifshitz (email: shayli@technion.ac.il) or to Z.K. (email: zhkang@suda.edu.cn)

Syngas, a mixture of $H_2$ and CO, is a critical feedstock for production of synthetic fuels and industrial chemicals via well-established industrial processes such as the Fischer–Tropsch process (commercialized by Sasol and Shell)[1, 2]. The $H_2$/CO ratio in syngas is of a great significance for meeting the requirements for specific products: $H_2$/CO = 2:1 for methanol and $H_2$/CO = 1:1 for dimethyl ether for example. The conventional production approach of syngas is based on reforming non-renewable fossil fuels (e.g., coal, petroleum coke, and natural gas)[3], which increases the consumption of fossil fuel and aggravates the energy crisis. Synthesizing syngas with a controlled $H_2$/CO ratio by reduction of $CO_2$ not only contributes to the solution of the energy crisis, but at the same time reduces the amount of greenhouse gases ($CO_2$).

$CO_2$ reduction to CO and hydrogen evolution reactions (HER) per se are two independent major and important fields. Electrochemical (EC) and photoelectrochemical (PEC) methods integrating $CO_2$ reduction reaction and HER are key components of prospective technologies for renewable syngas[4]. Different types of semiconductors have been combined with an efficient catalyst for $CO_2$ reduction to produce syngas by the PEC approach[5–8]. Cu-ZnO/GaN/n⁺–p Si was recently reported as a highly efficient PEC catalyst to produce syngas with a tunable $H_2$/CO ratio (between 1:2 and 4:1)[5]. Metal or metal-based composites, including Ag[9], Cu[10], Ru(II) polypyridyl complex[11], Ag/$C_3N_4$[12], and Re-functionalized graphene oxide[13], have been investigated for EC reduction of $CO_2$ and $H^+$/$H_2O$ to syngas. The different methods used to tune the ratio of $H_2$/CO include altering the $CO_2$ flow rate[14] and pressure[15], the reaction temperature[16], and the applied potential[9]. Different crystalline sites of Au catalyze different reaction channels (edge sites initiate CO generation and corner sites $H_2$ generation)[17, 18]. A novel pulsed-bias technique using Cu

as the catalyst was recently applied to tune the $H_2$/CO ratio in syngas between ~32:1 and 9:16 by using different pulse times for the same working potential. The selectivity is however limited and $CH_4$ and $C_2H_4$ by-products affect the purity of syngas[10]. Using Ag/$C_3N_4$[12], the $H_2$/CO ratio in the produced syngas can be tuned from 100:1 to 2:1 by controlling the applied potential and the Ag loading on graphitic carbon nitride but the total current density is lower than 1 mA/cm² at −0.6 V. The previous EC attempts to synthesize renewable syngas are still characterized by an unsatisfactory performance including some of the following disadvantages: a high onset overpotential necessary to initiate the $CO_2$ reduction reaction, a low CO + $H_2$ generation current density, a small selectivity of CO production, and a poor stability of the generation current density and the Faradaic efficiency (FE) of $H_2$ and CO.

We hereby propose a design concept of a cheap composite EC catalyst for a tunable, stable, selective, and efficient production of syngas, made of three components: a HER catalyst, a $CO_2$ reduction catalyst toward CO, and a catalyst which stabilizes the active hydrogen (H•) necessary to trigger both HER and the $CO_2$ reduction reactions. For HER, we choose several known catalysts ($Co_3O_4$, Pt, $MoS_2$, and Au). For $CO_2$ reduction, we apply graphitic carbon nitride ($C_3N_4$) since carbon-bonded nitrogen groups including pyridinic N, pyrrolic N, and graphitic N have recently been proposed as active sites for $CO_2$ reduction to CO[19–21]. $C_3N_4$ has a porous structure and is shown as a good substrate for dispersion of catalytic nanoparticles[22–25]. The selected catalyst for stabilization of active hydrogen (H•) are carbon dots (CDots)[26, 27], which possess significant adsorption capabilities for $H^+$[28, 29] and $CO_2$[30] and exhibit excellent ability of electron transfer[31, 32] necessary for H• generation ($H^+ + e^- \rightarrow H•$). CDots also improve the conductivity of $C_3N_4$.

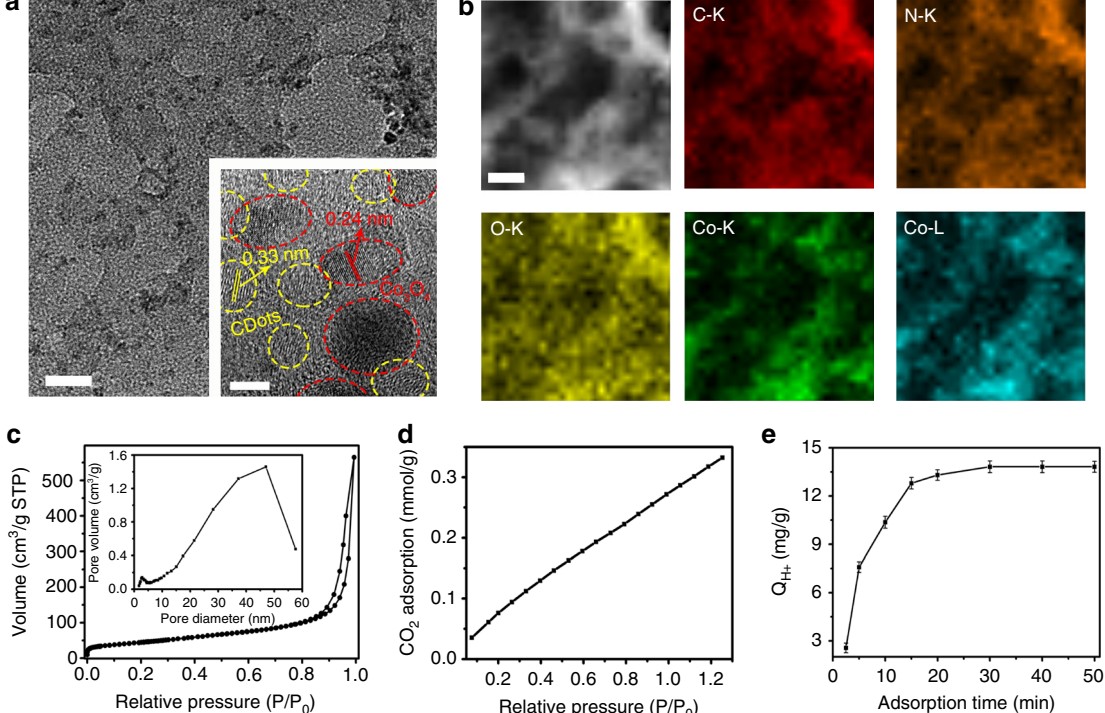

**Fig. 1** Characterization of the $Co_3O_4$-CDots-$C_3N_4$. **a** TEM image of a grain of the $Co_3O_4$-CDots-$C_3N_4$ and a HRTEM image of $Co_3O_4$-CDots-$C_3N_4$ (inset), scale bar 20 nm and 5 nm (inset). **b** STEM micrograph and the corresponding elemental mapping of C-K, N-K, O-K, Co-K, and Co-L for the $Co_3O_4$-CDots-$C_3N_4$, scale bar 50 nm. **c** $N_2$ adsorption–desorption isotherm and the corresponding pore-size distribution of $Co_3O_4$-CDots-$C_3N_4$ (inset). **d** $CO_2$ adsorption isotherm of $Co_3O_4$-CDots-$C_3N_4$. **e** The time-course adsorption of $H^+$ by the $Co_3O_4$-CDots-$C_3N_4$. The adsorption is 13.8 mg/g. Experiments were performed in triplicates and results are shown as mean ± standard deviation

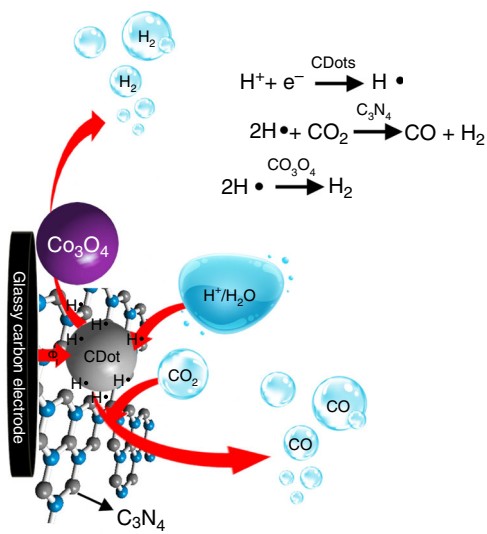

**Fig. 2** Schematic diagram of the reaction mechanism induced by $Co_3O_4$-CDots-$C_3N_4$. CDots are the generation site of H•, $Co_3O_4$ the generation site of $H_2$, and $C_3N_4$ the generation site of CO

Here we found that the $Co_3O_4$-CDots-$C_3N_4$ is the best EC catalyst for syngas production studied within the present work. We thus first describe the structural characterization of this catalyst. We then present the design concept of the three-component catalyst for syngas production and detail its working mechanisms. We follow by showing that this concept is valid and that $Co_3O_4$-CDots-$C_3N_4$ is indeed an efficient, tunable, stable, selective, and cheap EC catalyst for syngas production. It initiates $CO_2$ reduction to CO in aqueous solutions at a low overpotential (0.17 V vs. reversible hydrogen electrode (RHE)) and the total current density of $H_2$ and CO generation may reach up to 15 mA/cm² at a potential of −1.0 V vs. RHE. The $H_2$/CO ratio of syngas generated by $Co_3O_4$-CDots-$C_3N_4$ is tunable from 0.07:1 to 4:1 by controlling the applied potential. Dedicated experiments highlight the different mechanisms by which syngas production applying $Co_3O_4$-CDots-$C_3N_4$ is controlled and manipulated. Finally, the generality of the catalyst design concept applying Pt, $MoS_2$, and Au as the HER catalyst is demonstrated. The significance of this work is thus twofold. First, it presents $Co_3O_4$-CDots-$C_3N_4$ as an efficient, cheap, tunable, and stable catalyst for syngas production on one hand. Second, it provides an avenue to design catalysts for controlled and tuned production of syngas in particular and other chemicals of interest to the chemical industry in general.

## Results

**Characterization of $Co_3O_4$-CDots-$C_3N_4$.** Transmission electron microscopy (TEM) image (Fig. 1a) reveals that the $Co_3O_4$-CDots-$C_3N_4$ consist of nm-sized $Co_3O_4$ nanoparticles (NPs) and CDots dispersed on the $C_3N_4$ matrix. The two different types of nanocrystals shown in the inset of Fig. 1a are identified as $Co_3O_4$ NPs with a d-spacing of 0.24 nm consistent with $Co_3O_4$ (311)[33] and CDots with an interplanar spacing of 0.33 nm (see Supplementary Fig. 1)[34]. Figure 1b displays a scanning transmission electron microscopy (STEM) micrograph and its corresponding chemical maps of C-K, N-K, O-K, Co-K, and Co-L for the $Co_3O_4$-CDots-$C_3N_4$. Co-K, Co-L, and O-K cover the entire $C_3N_4$ area monitored, indicating that $Co_3O_4$ NPs are evenly distributed on the $C_3N_4$ sheet. X-ray diffraction (XRD) shows (Supplementary Fig. 2) the typical diffraction lines of $Co_3O_4$ and $C_3N_4$. Besides the average size of $Co_3O_4$ NPs derived from the XRD spectrum of the $Co_3O_4$-CDots-$C_3N_4$ using the Debye–Scherrer equation (Supplementary Table 1), is about 10 nm, which is consistent with the

size of $Co_3O_4$ NPs in the TEM image. Energy dispersive X-ray absorption (EDX) analysis of the $Co_3O_4$-CDots-$C_3N_4$ (Supplementary Fig. 3) reveals an elemental atom composition of C (46.65 at. %), N (48.74 at. %), O (3.01 at. %), and Co (1.60 at. %). X-ray photoelectron spectroscopy (XPS) of the composite (Supplementary Fig. 4) shows C 1s, N 1s, O 1s, and Co 2p peaks with a similar elemental composition to that deduced by EDX (C (39.9 at. %), N (52.5 at. %), O (5.8 at. %), and Co (1.8 at. %)) (Supplementary Table 2). The Co $2p_{3/2}$ peak was fitted (Supplementary Fig. 4e) by using two synthetic peaks positioned at binding energy (BE) = 780.0 and 781.3 eV ($Co^{3+}$ and $Co^{2+}$, respectively) and the Co $2p_{1/2}$ peak was fitted (Supplementary Fig. 4e) by using two synthetic peaks positioned at BE = 795.0 and 797.3 eV ($Co^{3+}$ and $Co^{2+}$, respectively). Figure 1c shows type IV $N_2$ adsorption–desorption isotherms of the $Co_3O_4$-CDots-$C_3N_4$ with a H3-type hysteresis loop[35], indicating the formation of a porous structure (the pore-size distribution was derived from the isotherms; for more details, see Supplementary Methods) is presented in Fig. 1c inset; specific surface area ~160 m²/g). The porous $Co_3O_4$-CDots-$C_3N_4$ was found to adsorb a large amount of $CO_2$ (~0.33 mmol/g at 1.2 atm, Fig. 1d). This strong $CO_2$ adsorption capability is mainly due to $CO_2$ molecules adsorbed on the porous $C_3N_4$ surfaces, which constitute more than 90 wt.% of the composite catalyst (Supplementary Fig. 5a). It is nevertheless enhanced in a synergistic way by the incorporation of CDots (Supplementary Fig. 5b shows that the $CO_2$ adsorption on $Co_3O_4$-CDots-$C_3N_4$ and on CDots-$C_3N_4$ is about the same, while that on $Co_3O_4$-$C_3N_4$ is much lower). The adsorption of $H^+$ (Fig. 1e) is very rapid in the first 15 min, then it becomes slower with contact time, stopping additional absorption after 30 min. The amount of adsorbed $H^+$ (for more details, see Supplementary Methods) on $Co_3O_4$-CDots-$C_3N_4$ and CDots-$C_3N_4$ is about 13.8 mg/g, while the amount of adsorbed $H^+$ is only 5.11 mg/g for the $Co_3O_4$-$C_3N_4$ (Supplementary Fig. 6b). The incorporation of CDots thus significantly increases the adsorbed capacity of $H^+$. All these results indicate that the present electrocatalyst has a strong adsorption capacity for both $H^+$ and $CO_2$, which is significantly important for both processes of $CO_2$ reduction reaction and HER.

**The electrocatalyst design and reaction mechanism.** We first present the design concept of the ternary electrocatalyst and highlight its operation mode to produce syngas in a controllable, tunable way. The proposed electrocatalyst consists of three parts: an electrocatalyst for HER (e.g., $Co_3O_4$), an electrocatalyst for $CO_2$ reduction to CO (e.g., $C_3N_4$), and an electrocatalyst which triggers both reaction channels (e.g., CDots by trapping $H^+$ and $e^-$ and generating H•). A schematic diagram of the $Co_3O_4$-CDots-$C_3N_4$ ternary electrocatalyst syngas generation is presented in Fig. 2. CDots first trap $H^+$ from the solution and $e^-$ from the glassy carbon electrodes and combine them to form H•. The two other catalyst components compete for these H• species. They may diffuse to $C_3N_4$ (the $CO_2$ reduction catalyst due to its different N active sites[19–21]) and reduce $CO_2$ to CO. Alternatively they may diffuse to $Co_3O_4$ (the HER catalyst) and produce $H_2$. We thus suggest that CDots are the generation site of H•, $C_3N_4$ is the generation site of CO, and $Co_3O_4$ is the generation site of $H_2$. Syngas with tunable $H_2$/CO ratio can be thus achieved by balancing the $CO_2$ reduction channel and the HER channel. The different catalyst components have additional functions. Introduction of CDots to the composites enhances the adsorption of both $CO_2$ and $H^+$. Incorporation of CDots in the electrocatalyst improves its conductivity (Supplementary Fig. 7). $C_3N_4$ serves as the highly porous substrate in which the CDots and $Co_3O_4$ nanoparticles are incorporated. $C_3N_4$ thus provides a large surface area which

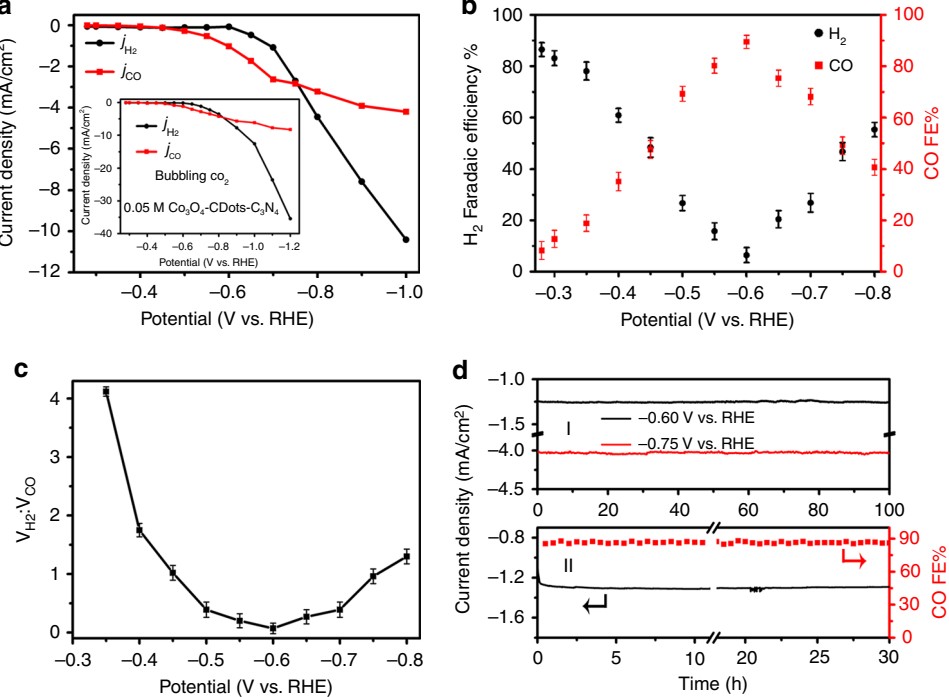

**Fig. 3** Electroreduction of $CO_2$ and $H^+/H_2O$ to syngas with adjustable $H_2/CO$ ratio. **a** The current density for HER ($j_{H2}$, black trace) and for $CO_2$ reduction, ($j_{CO}$, red trace) vs. the applied potential, catalyzed by $Co_3O_4$-CDots-$C_3N_4$ in $CO_2$-saturated 0.5 M $KHCO_3$ electrolyte. The inset shows the same experiment conducted with bubbling of $CO_2$ to the solution to overcome $CO_2$ consumption during the experiment. **b** The FEs of the reduction of $CO_2$ to CO (red points) and $H^+$ to $H_2$ (black points) catalyzed by $Co_3O_4$-CDots-$C_3N_4$ vs. the applied potential. Experiments were performed in triplicates and results are shown as mean ± standard deviation. **c** The volume ratio between $H_2$ and CO vs. the applied potential. The $H_2/CO$ volume ratio is about 1:1 at the potential of −0.45 V or −0.75 V vs. RHE. **d** The stability of the performance of $Co_3O_4$-CDots-$C_3N_4$ for producing syngas: operated at potentiostatic potential of −0.6 and −0.75 V for 100 h (**dI**); operated at potentiostatic potential of −0.6 V for 30 h (**dII**): current density vs. time (left axis) and FEs of CO production vs. time (right axis)

enhances the reaction activity and also guarantees a proximity between the different generation sites (CDots for H• generation, $C_3N_4$ for CO generation, and $Co_3O_4$ for $H_2$ generation). This proximity is essential for an efficient reaction. Experimental evidence substantiating this proposed mechanism will be given in the following sections.

**The electrocatalytic performance of $Co_3O_4$-CDots-$C_3N_4$.** We now show that the suggested design concept of the $Co_3O_4$-CDots-$C_3N_4$ indeed provides a tunable and stable production of syngas. The electrocatalytic performance of $Co_3O_4$-CDots-$C_3N_4$ for syngas production was tested in an airtight three electrodes electrochemical H-type cell. The gaseous reduction products monitored by a gas chromatography (GC) system were CO and $H_2$ while no other reduction liquid products were found by $H^1$ NMR. Figure 3a shows the current densities of CO ($j_{CO}$, red trace) and $H_2$ ($j_{H2}$, black trace) vs. different potentials acquired in a $CO_2$-saturated 0.5 M $KHCO_3$ (pH = 7.2) solution. The curves indicate a significant generation of both CO and $H_2$ but the $j_{CO}/j_{H2}$ ratio varies with the potential applied. The $j_{CO}$ curve shows that the $CO_2$ electrocatalytic reduction starts at an initial potential of −0.28 V vs. RHE (all potentials reported throughout this paper are with respect to this reference). This operating voltage corresponds to a low overpotential of 0.17 V (the equilibrium potential of $CO_2$/CO is at −0.11 V). Up to −0.45 V, both $j_{CO}$ and $j_{H2}$ are very small and above −0.45 V both currents increase. Between −0.45 and −0.75 V, $j_{CO}$ is larger than $j_{H2}$, indicating that the $CO_2$ reduction reaction predominates. For potentials larger than −0.75 V, the HER channel becomes more significant than the $CO_2$ reduction reaction. The increase of $j_{H2}$ with potential is about

constant (the $j_{H2}$ curve is a straight line). In contrast, the slope of the $j_{CO}$ curve decreases with the potential applied. We attribute this behavior to the consumption of $CO_2$ at high CO generation rates due to its low solubility or slow mass transfer. Indeed (inset of Fig. 3a), bubbling $CO_2$ into the electrolyte increases $j_{CO}$ but the increase rate of $j_{CO}$ with potential is still lower than that of $j_{H2}$. Figure 3b shows the FEs of CO and $H_2$ generation vs. the applied cathodic potentials. The FE of CO generation increases with the applied potential and reaches a maximum of 89% at −0.6 V. For higher potential, it decreases to ~40% with increasing potential to −0.8 V, most probably due to the limited mass transport of $CO_2$ in the electrolyte. In contrast, the FE of $H_2$ generation is about 86% for a low applied potential, decreases with potential to about 5% at −0.6 V and then rises again reaching 55% at −0.8 V. The total FE of CO and $H_2$ combined reaches up to 95%. Consequently, syngas with different $H_2/CO$ ratios can be obtained by altering the applied potential as shown in Fig. 3c. The volume ratio between $H_2$ and CO can be tuned from 4:1 to 0.07:1 and syngas with $H_2/CO = 1:1$ can directly be generated at the potentials of −0.45 V (total current density is 0.25 mA/cm²) and −0.75 V (total current density is 5.78 mA/cm²). Note that the catalyst mass activity for syngas generation reaches a high value of 10 A/$g_{catalyst}$ at a potential of −0.6 V. The stability of the $Co_3O_4$-CDots-$C_3N_4$ catalyst for electrocatalytic $CO_2$ reduction and HER (Fig. 3d) was studied as well. A negligible decay in current density was observed when operating the system for 100 h at the applied potential of both −0.6 V (Fig. 3dI, black trace, $CO_2$ reduction reaction predominates) and −0.75 V (Fig. 3dI, red trace, HER and $CO_2$ reduction are equal). It reveals that the $Co_3O_4$-CDots-$C_3N_4$ maintains high electrocatalytic stability during the 100-h test. An additional experiment validated the stability of

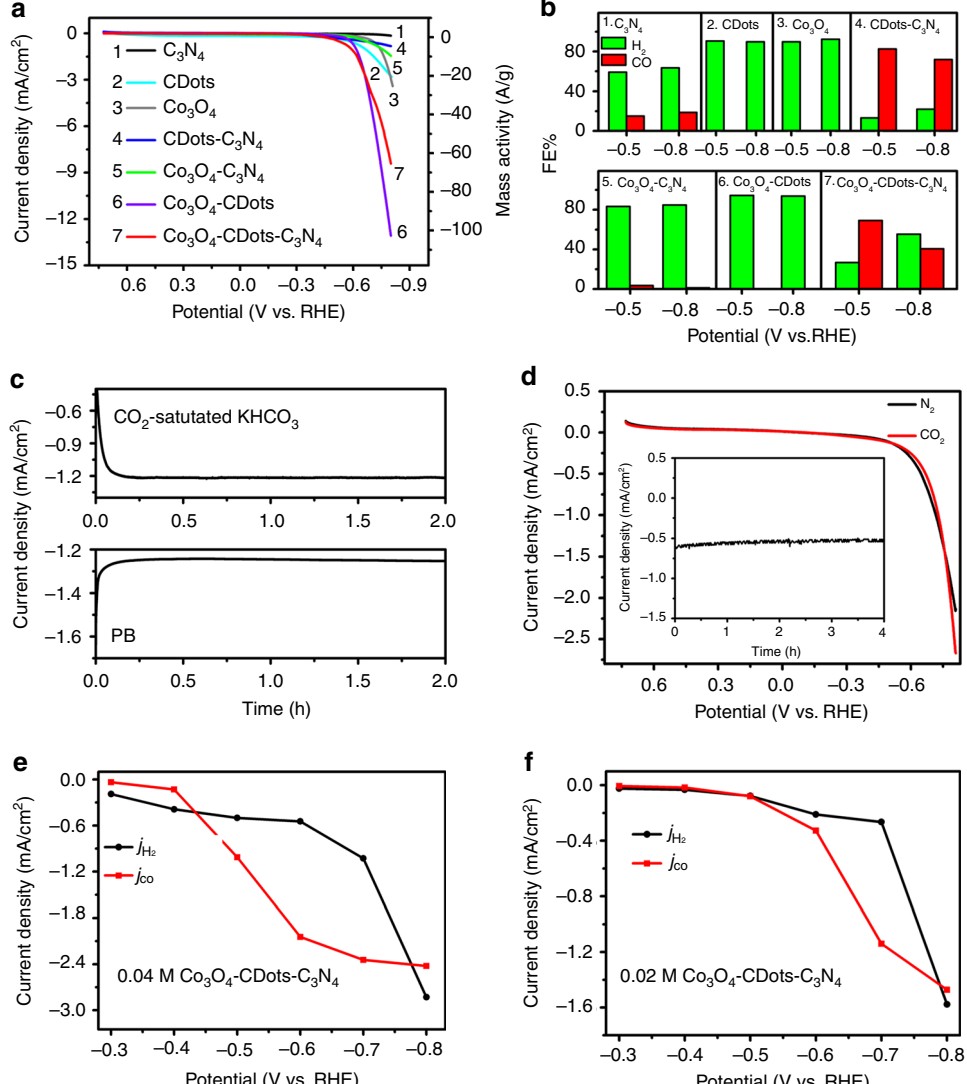

**Fig. 4** Electrochemical experiments providing insight into the electrochemical processes. **a** The linear sweep voltammetry curves (LSVs) for $C_3N_4$ (black trace, curve 1), CDots (cyan trace, curve 2), $Co_3O_4$ (gray trace, curve 3), CDots-$C_3N_4$ (blue trace, curve 4), $Co_3O_4$-$C_3N_4$ (green trace, curve 5), $Co_3O_4$-CDots (purple trace, curve 6), and $Co_3O_4$-CDots-$C_3N_4$ (red trace, curve 7) in $CO_2$-saturated 0.5 M $KHCO_3$ electrolyte, 10 mV/s. Current density on left y axis and mass activity on right y axis. The comparison of the curves allows determination of the role of the different catalyst components in the electrochemical reactions. **b** The FEs of the reaction products at −0.5 and −0.8 V, using $C_3N_4$, CDots, $Co_3O_4$, CDots-$C_3N_4$, $Co_3O_4$-$C_3N_4$, $Co_3O_4$-CDots, and $Co_3O_4$-CDots-$C_3N_4$, respectively, as catalysts. Note that only the $C_3N_4$ and CDots-$C_3N_4$ composites produce significant amounts of CO. **c** Total current density vs. time curves of $CO_2$ reduction to CO and HER at the potential of −0.6 V in $CO_2$-saturated $KHCO_3$ solution (0.5 M, pH = 7.2) and phosphate buffer (PB) solution (pH = 7.2), respectively. **d** LSVs for the $Co_3O_4$-CDots-$C_3N_4$ in $N_2$- (black trace) and $CO_2$-saturated (red trace) MeCN containing 0.5 M [BMIM]$PF_6$, 10 mV/s. The inset shows the total current density vs. time curves of $CO_2$ reduction at the potential of −0.6 V in $CO_2$-saturated MeCN containing 0.5 M [BMIM]$PF_6$. The current in the inset is much smaller than in **c** and no reaction products are detected indicating that $H^+$ is essential for both HER and $CO_2$ reduction to CO. **e** Current density for HER ($j_{H_2}$, black trace) and current density for $CO_2$ reduction, ($j_{CO}$, red trace) vs. the applied potential, catalyzed by 0.04 M $Co_3O_4$-CDots-$C_3N_4$ in a $CO_2$-saturated 0.5 M $KHCO_3$ electrolyte. **f** Current density for HER ($j_{H_2}$, black trace) and current density for $CO_2$ reduction, ($j_{CO}$, red trace) vs. the applied potential, catalyzed by 0.02 M $Co_3O_4$-CDots-$C_3N_4$ in $CO_2$-saturated 0.5 M $KHCO_3$ electrolyte; note that the reduction of the amount of the HER catalyst component shifts the balance of gas generation toward enhanced CO generation

both the current density and the FE for 30 h (it is very likely that the FE is stable for 100 h as well).

**Investigation of the catalytic mechanisms**. A series of controlled experiments were carried out to validate and further understand the catalytic mechanism operating with the $Co_3O_4$-CDots-$C_3N_4$ proposed in a previous section (Fig. 2). Issues investigated included: the generation sites of H•, CO, and $H_2$; the role of $H^+$; the roles of CDots, $C_3N_4$, and $Co_3O_4$; the role of the proximity between the CDots, $C_3N_4$, and $Co_3O_4$; ways in which the relative

significance of the different reaction channels and the resulting CO/$H_2$ volume ratio can be manipulated; other HER-CDots-$C_3N_4$ systems with different HER materials. We first studied the linear sweep voltammetry (LSV) curves of different combinations of the three components comprising the ternary $Co_3O_4$-CDots-$C_3N_4$. The comparison of the different LSV curves (Fig. 4a) and the complementary study of the CO and $H_2$ composition generated during the electrocatalytic processes catalyzed by the different components (Fig. 4b) enabled a clear determination of both the generation sites of the CO and $H_2$ and the role of the three basic components used. We first discuss the LSVs showing the apparent

current density (current density per geometrical area), which is also equivalent to the mass activity of the catalysts (current per gram catalyst) as shown in Fig. 4a. Further on, we also give the corresponding turn on frequencies (TOFs) and the real current densities derived from the real surface areas of the components of the different combinations of catalysts in Fig. 4a. LSVs of $C_3N_4$ (Fig. 4a, curve 1, black trace, morphology of $C_3N_4$ is shown in Supplementary Fig. 8a), CDots (Fig. 4a, curve 2, cyan trace, morphology of CDots is shown in Supplementary Fig. 1), $Co_3O_4$ (curve 3, gray trace, morphology of $Co_3O_4$ is shown in Supplementary Fig. 8b), CDots-$C_3N_4$ (curve 4, blue trace, morphology of CDots-$C_3N_4$ is shown in Supplementary Fig. 8c), $Co_3O_4$-$C_3N_4$ (curve 5, green trace, morphology of $Co_3O_4$-$C_3N_4$ is shown in Supplementary Fig. 8d), $Co_3O_4$-CDots (curve 6, purple trace, morphology of $Co_3O_4$-CDots is shown in Supplementary Fig. 8e), and $Co_3O_4$-CDots-$C_3N_4$ (curve 7, red trace) were performed using a $CO_2$-saturated 0.5 M $KHCO_3$ (pH = 7.2) solution (Fig. 4a). The LSVs of $C_3N_4$ (curve 1, black trace), CDots (curve 2, cyan trace), $Co_3O_4$ (curve 3, gray trace), CDots-$C_3N_4$ (curve 4, blue trace), and $Co_3O_4$-$C_3N_4$ (curve 5, green trace) show poor electrocatalytic performances evident by their low current densities and high onset potentials (Fig. 4a). Only the $Co_3O_4$-CDots (curve 6) and the $Co_3O_4$-CDots-$C_3N_4$ (curve 7) exhibit high current densities and low onset potentials (Fig. 4a). CDots generate $H_2$ only, but not CO (Fig. 4b). We explain it by their trapping and stabilizing H•, which may generate a small amount of $H_2$ even in the absence of a HER catalyst. The LSV of $Co_3O_4$ (curve 3, gray trace) shows a relatively small activity producing only $H_2$ (Fig. 4b). In comparison, $Co_3O_4$-CDots (curve 6, purple trace) produces a much larger amount of $H_2$ compared to pure CDots or pure $Co_3O_4$. We attribute it to the effect of the CDots, which greatly enhance the electrocatalytic performance of $Co_3O_4$ by providing H•, which is necessary for generation of $H_2$. We thus conclude that $Co_3O_4$ is the $H_2$ generation site while CDots are the generation site of H• and both are needed for a large generation rate of $H_2$. Now, we prove that the generation site of CO is $C_3N_4$. Figures 3a, b and 4a, b indicate that significant amounts of CO are generated by either the CDots-$C_3N_4$ or the $Co_3O_4$-CDots-$C_3N_4$. $Co_3O_4$ was shown to generate $H_2$ only, which leaves CDots-$C_3N_4$ as the producer of CO. CDots per se produce only $H_2$ as previously discussed. They however are essential for CO generation by CDots-$C_3N_4$ since they significantly enhance the $CO_2$ adsorption, adsorb $H^+$, and stabilize H•. $C_3N_4$ per se catalyzes only a very small (negligible) current (Fig. 4a, curve 1) so that the very little amount of gas produced by pure $C_3N_4$ (Fig. 4b) contains more $H_2$ than CO. The supply of H• by the CDots is necessary to promote the generation of CO in $C_3N_4$. Figure 4a shows that each catalyst component adds to the activity by enhancing one of the three reactions (H• generation, $H_2$ generation, and CO generation) but the complete three components composite $Co_3O_4$-CDots-$C_3N_4$ is necessary for intense generation of syngas. We further studied the effect of the type of mixing of the different catalyst components on the catalytic activity. The LSVs of physical mixtures of catalysts ($Co_3O_4$ + $C_3N_4$, CDots + $C_3N_4$, and $Co_3O_4$ + CDots + $C_3N_4$) were compared to those of composites of the same components chemically blended ($Co_3O_4$-$C_3N_4$, CDots-$C_3N_4$, and $Co_3O_4$-CDots-$C_3N_4$). The chemically prepared composites had much larger activities (current densities) than their corresponding physical mixtures (Supplementary Fig. 9). We attribute the large activity of the chemically prepared composites to the close proximity between the different catalysts (active sites) in the composite materials. In contrast, physical mixing does not provide such a proximity so that the large distance between the active sites hinders the activity of the physically mixed catalysts.

The comparison between the LSVs of the different catalyst components and their composites (Fig. 4a) should be done

carefully. The catalyst areal density was kept the same (0.127 mg/$cm^2$) for all (3 mm in diameter) electrodes. Since the composition of the $Co_3O_4$-CDots-$C_3N_4$ was 6 wt% $Co_3O_4$, 1 wt% CDots, and 93 wt% $C_3N_4$, it follows that the amount of CDots or $Co_3O_4$ in the CDots, $Co_3O_4$, or $Co_3O_4$-CDots electrodes was much larger than in the $C_3N_4$ containing composite electrodes. This means that the reaction activity per catalyst mass of curves 2, 3, and 6 is very low compared to that of the $Co_3O_4$-CDots-$C_3N_4$ electrode (Fig. 4a, curve 7). Since we deal with three different catalysts with different functions, we should calculate the mass activity (current per gram catalyst) for each component separately. This was done in Supplementary Fig. 10. The reaction activity per mass of a single catalyst component ($Co_3O_4$ or CDots) of $Co_3O_4$-CDots (Fig. 4a, curve 6), e.g., is actually lower by more than an order of magnitude than the activity per mass of a single catalyst component of that of the $Co_3O_4$-CDots-$C_3N_4$ catalyst (Fig. 4a, curve 7) though curve 6 appears to indicate (Fig. 4a) a larger activity than curve 7 (compare Fig. 4a to Supplementary Fig. 10). Supplementary Fig. 10 clearly shows that starting from a single catalyst component, the addition of each of the two other components increases the activity of syngas production, i.e., all components are necessary for an optimized syngas generation. Similarly, we calculate the turn on frequencies (TOFs) of the catalysts compositions of Fig. 4a. TOF = (number of reacted electrons per time/number of catalyst active sites). We approximate the number of active sites by the number of catalyst atoms. Since we have three different catalyst components with three different functions, we calculate the TOFs per each catalyst component (CDots, $Co_3O_4$, and $C_3N_4$). Supplementary Fig. 11 shows (similar to Supplementary Fig. 10) that the addition of each single component increases the syngas production activity and all components are necessary for the optimal performance. We further measure the BET and the electrochemical real surface areas of the catalyst components of each combination shown in curves 1–7 (Supplementary Tables 3–4). We use the electrochemical surface areas (ECSAs) to calculate the real current densities related to the specific catalyst components that constitute the seven combinations shown in curves 1–7. Supplementary Fig. 12 shows that similar to the mass activities and the TOFs curves, the real current densities of one component increase with the addition of a second component and are the largest when all three catalyst components combine to a three-component catalyst composite. Figure 3 shows that the ternary concept design indeed yields an optimal performance of its different components balancing between $CO_2$ reduction and HER. This explains the high value of 10 A/$g_{catalyst}$ obtained at −0.6 V. The $CO_2$ and $H^+$ adsorption measurements (Supplementary Figs. 5 and 6) of the individual different catalyst components (CDots, $Co_3O_4$, and $C_3N_4$) and their composites (CDots-$C_3N_4$, $Co_3O_4$-$C_3N_4$, CDots-$Co_3O_4$, and $CO_3O_4$-CDots-$C_3N_4$) reveal another effect which improves the ternary composite catalyst performance. The synergism of the three components acting simultaneously enhances the adsorption of the ternary composite by a factor of 2–3 with respect to the adsorption of the individual components. Real surface area measurements (BET and electrochemical) of the catalyst compositions (Supplementary Table 3) indicate that the inclusion of the $C_3N_4$ component results in a high surface area (($S_{real}/S_{geometrical}$) = 200 for BET and 30 for electrochemical) while it is lower by an order of magnitude for the nanoparticle catalysts (CDots, $Co_3O_4$, or CDots-$Co_3O_4$). The dispersion of the catalysts nanoparticles on the $C_3N_4$ surface seems to explain the synergistic adsorption behavior.

To study the role of $H^+$ in syngas generation, we investigated the electrocatalytic performance of $Co_3O_4$-CDots-$C_3N_4$ in aqueous solutions with the same pH value (HER in phosphate buffer solution, pH = 7.2; syngas reaction in $CO_2$-saturated 0.5 M

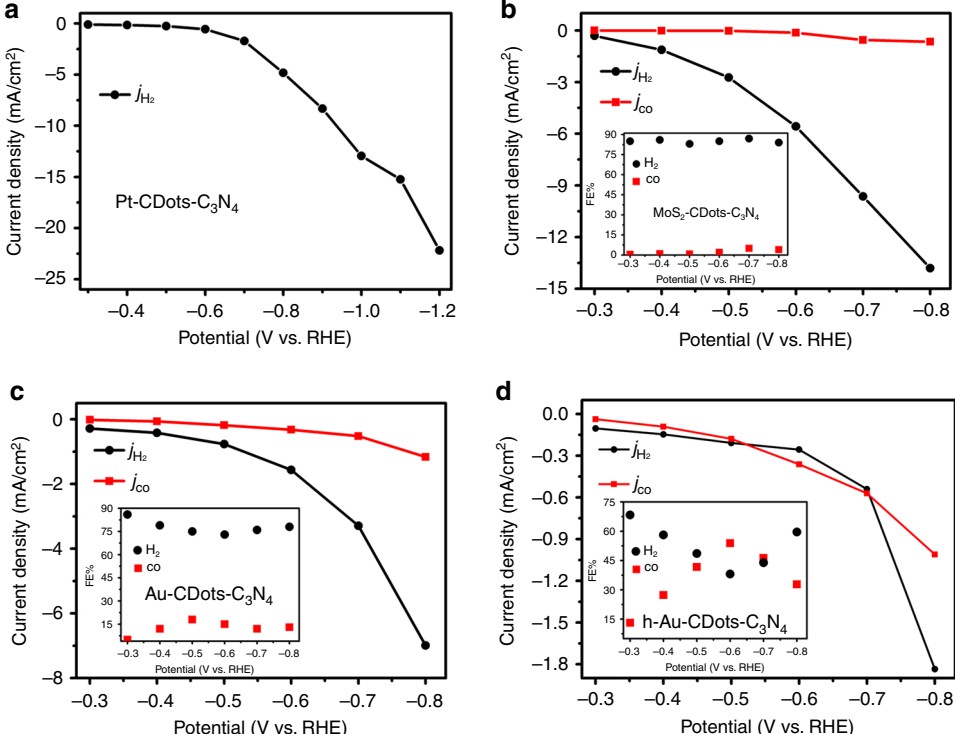

**Fig. 5** Catalytic activity of composite catalysts with different HER catalysts. **a** Current density for HER ($j_{H_2}$, black trace) vs. the applied potential, catalyzed by Pt-CDots-$C_3N_4$ in $CO_2$-saturated 0.5 M $KHCO_3$ electrolyte. **b** Current density for HER ($j_{H_2}$, black trace) and current density for $CO_2$ reduction to CO, ($j_{CO}$, red trace) and FEs of $H_2$ and CO (inset) vs. the applied potential, catalyzed by MoS$_2$-CDots-$C_3N_4$ in $CO_2$-saturated 0.5 M $KHCO_3$ electrolyte. **c** Current density for HER ($j_{H_2}$, black trace) and current density for $CO_2$ reduction, ($j_{CO}$, red trace) and FEs of $H_2$ and CO (inset) vs. the applied potential, catalyzed by Au-CDots-$C_3N_4$ in $CO_2$-saturated 0.5 M $KHCO_3$ electrolyte. **d** Current density for HER ($j_{H_2}$, black trace) and current density for $CO_2$ reduction, ($j_{CO}$, red trace) and FEs of $H_2$ and CO (inset) vs. the applied potential, catalyzed by Au-CDots-$C_3N_4$ in $CO_2$-saturated 0.5 M $KHCO_3$ electrolyte. The amount of Au in **d** is half of that in **c** (marked as h-Au-CDots-$C_3N_4$) leading to an increase of the CO/$H_2$ ratio in **d** with respect to that in **c**

$KHCO_3$ solution, pH = 7.2) and in an ionic liquid (without $H^+$). The current density–time curves of pure HER and syngas reactions at the potential of −0.6 V (at −0.6 V the syngas reaction produces ~90% CO) are shown in Fig. 4c. A stable current density (~−1.21 mA/cm$^2$) for syngas is observed at −0.6 V while −1.25 mA/cm$^2$ for HER is reached at the same potential. Then, ionic liquid was used as the electrolyte solution to eliminate $H^+$. The electrocatalytic performance of $Co_3O_4$-CDots-$C_3N_4$ was tested at the potential of −0.6 V in a $CO_2$-saturated MeCN solution containing 0.5 M [BMIM]PF$_6$. The current density is only about 0.61 mA/cm$^2$ in a $CO_2$-saturated ionic liquid (Fig. 4d), which is much lower than that obtained for $CO_2$ reduction in $CO_2$-saturated $KHCO_3$ solution (−1.21 mA/cm$^2$). Notably, no gaseous products of CO reduction could be detected suggesting that $H^+$ plays an important role for $CO_2$ reduction to CO in the present $Co_3O_4$-CDots-$C_3N_4$ catalyst system.

It was shown that the $H_2$/CO volume ratio obtained using the $Co_3O_4$-CDots-$C_3N_4$ catalyst system is determined by the balance between the HER channel and the $CO_2$ reduction channel to CO. This ratio was tuned by modifying the potential (Fig. 3). Another plausible tuning method is the decrease of the amount of the HER catalyst component ($Co_3O_4$) thus reducing the HER activity and increasing the $CO_2$ reduction activity. We therefore measured the electrocatalytic activities of $Co_3O_4$-CDots-$C_3N_4$ produced using lower amounts of Co (0.04 and 0.02 M $Co_3O_4$ loadings with respect to the standard one of the present work (0.05 M)). Figures 3a and 4e, f show the $j_{H_2}$ and $j_{CO}$ obtained at different applied potentials for 0.05, 0.04, and 0.02 M $Co_3O_4$-CDots-$C_3N_4$, respectively. It is obvious that the CO/$H_2$ ratio was strongly affected by the change of the $Co_3O_4$ HER catalyst amount.

Finally, the design concept of the HER-CDots-$C_3N_4$ ternary catalyst was applied for three additional HER active electrocatalysts: Pt, MoS$_2$, and Au to form Pt-CDots-$C_3N_4$, MoS$_2$-CDots-$C_3N_4$, and Au-CDots-$C_3N_4$. The structural characterization of these catalysts is given in Supplementary Fig. 13. The electrocatalytic activity of these three ternary composite catalysts for producing syngas were tested under the same conditions as for $Co_3O_4$-CDots-$C_3N_4$. Applying the Pt-CDots-$C_3N_4$ electrocatalyst (Fig. 5a), only hydrogen was detected in the gas phase and no reduction products from $CO_2$ reduction were observed. Pt is considered as the most efficient electrocatalyst to facilitate HER[36]. The application of Pt-CDots-$C_3N_4$ as a catalyst shifts the balance between HER and $CO_2$ reduction toward $H_2$ generation, increases the intensity of the efficient HER channel and completely suppresses the $CO_2$ reduction to CO. The MoS$_2$-CDots-$C_3N_4$ electrocatalyst is still sufficiently HER active to produce close to 90% $j_{H_2}$, but a small amount of $j_{CO}$ (a few percent of the total current) is nevertheless observed. The FEs of CO are no more than 10% for the MoS$_2$-CDots-$C_3N_4$ (inset of Fig. 5b). The Au-CDots-$C_3N_4$ electrocatalyst (Fig. 5c) still exhibits a higher HER activity than the $CO_2$ reduction activity, but the amount of CO is much larger than for the MoS$_2$-CDots-$C_3N_4$ catalyst (the FE of CO production reaches 25%, inset of Fig. 5c). It can be thus concluded that the concept of the ternary HER-CDots-$C_3N_4$ is general and valid for HER catalysts different than $Co_3O_4$. Achievement of a relatively high amount of CO/$H_2$ however requires the application of a HER catalyst with only a medium activity pushing the balance between HER and $CO_2$ reduction toward $CO_2$ reduction. The smaller the HER activity, the larger relative amount of CO is obtainable. This conclusion

was directly checked by reducing the amount of Au in the ternary Au-CDots-C$_3$N$_4$ electrocatalyst. The electrochemical tests (Fig. 5d) show that decreasing the amount of Au by a factor of two enabled the generation of syngas with a CO to H$_2$ volume ratio larger than one. A striking property of the h-Au-CDots-C$_3$N$_4$ catalyst is its extremely high mass activity for producing syngas, i.e., >700 A/g$_{Au}$ for CO production and >700 A/g$_{Au}$ H$_2$ production (~1500 A/g$_{Au}$ for the total current) at −0.7 V for the catalyst shown in Fig. 5d. This activity is two orders of magnitude larger than previously reported for efficient Au electrodes for CO production[17, 18]. This high mass activity allows a reduction of the electrode cost when precious catalysts (e.g., Au) are applied.

## Discussion

The design concept of the HER-CDots-C$_3$N$_4$ EC catalyst for syngas generation was introduced and its electrocatalytic performance for syngas production in aqueous solutions was studied, applying Co$_3$O$_4$, MoS$_2$, Au, and Pt as the HER catalyst component. The Co$_3$O$_4$-CDots-C$_3$N$_4$ electrocatalyst was found most efficient for syngas production among the composite combinations investigated. The Co$_3$O$_4$-CDots-C$_3$N$_4$ is capable of controlling the balance between the HER channel and CO$_2$ reduction channel. The Co$_3$O$_4$-CDots-C$_3$N$_4$ initiates the reaction of CO$_2$ reduction to CO in aqueous solutions at a low overpotential (0.17 V) while the total current density reaches up to 15 mA/cm$^2$ at a potential of −1.0 V. The mass activity of the Co$_3$O$_4$-CDots-C$_3$N$_4$ is ~10 A/g$_{catalyst}$ at −0.6 V when the total mass of the catalyst is considered and 1–2 orders of magnitude larger when the mass of the HER catalyst is considered (which is ~0.5–5% of that of the total catalyst weight). The Co$_3$O$_4$-CDots-C$_3$N$_4$ induces high FEs (95%) and is characterized by a stable production of syngas (over 100 h). Notably, the H$_2$/CO ratio of syngas produced applying Co$_3$O$_4$-CDots-C$_3$N$_4$ is tunable from 0.07:1 to 4:1 by controlling the applied potential. The H$_2$/CO may be also tuned by varying the amount of Co$_3$O$_4$ in the Co$_3$O$_4$-CDots-C$_3$N$_4$.

Dedicated experiments validated the catalyst design concept and provided additional insight to the syngas generation processes. C$_3$N$_4$ and Co$_3$O$_4$ are the activity sites for CO$_2$ reduction reaction and HER, respectively. CDots are the generation site of H• needed to trigger both the reduction of CO$_2$ to CO and the HER. The three-component catalyst concept is a general one and may be applied to a host of other materials. The versatility of the three components composite design may open a powerful pathway for the development of high-performance catalysts for syngas production as well as for other chemicals generation. Such an efficient and cost-effective electrocatalytic system has a high potential to be employed for the large-scaled production of syngas and controlled mixtures of other chemicals from CO$_2$.

## Methods

**Instruments**. Transmission electron microscopy (TEM), high-resolution transmission electron microscopy (HRTEM), and scanning TEM (STEM) images were obtained using a FEI/Philips Tecnai G2 F20 TWIN transmission electron microscope. The energy dispersive X-ray spectroscopy (EDS) analyses were taken on a FEI-quanta 200 scanning electron microscope with an acceleration voltage of 20 kV . The crystal structure of the resultant products was characterized by X-ray diffraction (XRD) using an X'Pert-ProMPD (Holland) D/max-γAX-ray diffractometer with Cu Kα radiation (λ = 0.154178 nm). X-ray photoelectron spectroscopy (XPS) was obtained by using a KRATOS Axis ultra-DLD X-ray photoelectron spectrometer with a monochromatized Mg Kα X-ray source (hν = 1283.3 eV). The electrocatalysis reactions were tested by a Model CHI 660C workstation (CH Instruments, Chenhua, Shanghai, China). The electrochemical impedance spectroscopy (EIS) measurements were obtained applying a CHI 832 electrochemical instrument (CHI Inc., USA).

**Materials**. KHCO$_3$ (99.7%) and Nafion perfluorinated resin solution (5 wt.%) were purchased from Sigma-Aldrich; hydrogen (99.999%), nitrogen (99.999%), and carbon dioxide (99.999%) were purchased from Airgas; Nafion®212 membrane was

purchased from Dupont; Toray Carbon Paper (TGP-H-60) was purchased from Alfa Aesar; All chemicals were purchased from Sigma-Aldrich unless specifically stated. Milli-Q ultrapure water (Millipore, ≥18 MΩ/cm) was used throughout the work.

**Fabrication of electrocatalysts**. CDots were synthesized by our previously reported electrochemical etching method[34]. After 30-days reaction, a dark yellow solution containing CDots was formed in the reaction cell. It was then purified and concentrated to form a CDots solution of 3 mg/mL. For C$_3$N$_4$ fabrication, 10 g of melamine powder was put into an alumina crucible with a cover and then heated to 550 °C at a rate of 0.5 °C per min in a muffle furnace and maintained at this temperature for 3 h. The yellow powder (C$_3$N$_4$) was obtained after cooling down to room temperature. Co$_3$O$_4$ NPs was synthesized by hydrothermal method. Twenty mL Co(NO$_3$)$_2$ (0.01 M) solution was added into an alumina crucible with a cover and then heated to 550 °C at a rate of 0.5 °C per min in a muffle furnace and maintained at this temperature for 3 h. The black powder (Co$_3$O$_4$ NPs) was obtained after cooling down to room temperature. For preparation of Co$_3$O$_4$-C$_3$N$_4$, CDots-C$_3$N$_4$ or Co$_3$O$_4$-CDots-C$_3$N$_4$, 10 g of melamine powder was mixed with 10 mL solution containing 0.05 M Co(NO$_3$)$_2$, CDots, CDots and 0.05 M Co(NO$_3$)$_2$, respectively. Then, the mixture was put into an alumina crucible with a cover and heated to 550 °C at a rate of 0.5 °C per min in a muffle furnace and maintained at this temperature for 3 h. For preparation of Co$_3$O$_4$-CDots, 10 mL solution containing 0.05 M Co(NO$_3$)$_2$ and CDots (3 mg/mL) was put into an alumina crucible with a cover and heated to 550 °C at a rate of 0.5 °C per min in a muffle furnace and maintained at this temperature for 3 h.

**Synthesis of Pt-CDots-C$_3$N$_4$**. An aliquot of 0.3 g CDots-C$_3$N$_4$ (obtained by heating melamine at 550 °C for 3 h) was added into 10 mL H$_2$PtCl$_6$ (2 mM) aqueous solution and stirred for 12 h. After centrifuging, the precipitate was irradiated by UV light for 10 h. The resulting product was collected by centrifugation and dried in a vacuum at 60 °C for 12 h.

**Synthesis of MoS$_2$-CDots-C$_3$N$_4$**. An aliquot of 0.3 g CDots-C$_3$N$_4$ (obtained by heating melamine at 550 °C for 3 h) was added into 15 mL aqueous solution containing Na$_2$MoO$_4$ (0.0625 g) and L-cysteine (0.1 g). The mixed solution was stirred 3 min. After that, the mixture was poured into a Teflon-lined stainless steel autoclave, and heated at 180 °C for 24 h. After the autoclave was cooled down to room temperature, the resulting black sediments were collected by centrifugation (10,000 rpm, 10 min) and washed with deionized water and ethanol for several times, and then dried in a vacuum oven at 80 °C for 12 h[37].

**Synthesis of Au-CDots-C$_3$N$_4$ and h-Au-CDots-C$_3$N$_4$**. An aliquot of 0.3 g CDots-C$_3$N$_4$ (obtained by heating melamine at 550 °C for 3 h) was added into 10 mL HAuCl$_4$ (2 mM or 1 mM) aqueous solution and stirred for 12 h. After centrifuging, the precipitate was irradiated by UV light for 2 h. The resulting product was collected by centrifugation and dried in vacuum at 60 °C for 12 h[38].

**Electrocatalysis activity test**. Electrocatalysis activity test experiments were performed using a standard three-electrode configuration. A platinum wire was used as an auxiliary electrode and a saturated calomel electrode (SCE) was used as a reference electrode. The working electrode was either a catalyst-modified carbon fiber paper electrode (CFPE for short, 0.7 cm × 0.7 cm), or a catalyst-modified glassy carbon disk electrode (GCE for short, 3.0 mm diameter). For product analysis and constant-potential electrolysis experiment, the CFPE working electrode was a catalyst-modified carbon fiber paper electrode (0.7 cm × 0.7 cm). The preparation of the CFPE working electrode is as follows. An aliquot of 1.3 mg of electrocatalyst was ground with 0.1 mg polyvinylidene fluoride (PVDF) with a few drops of 1-methyl-2-pyrrolidone (MP) added to the produced mixture. The mixture was added into 10 mL 0.5% Nafion solution. After sonication, 1 mL dispersed solution was dropped directly onto the two sides of a 0.7 cm × 0.7 cm carbon fiber paper (the two sides of the carbon paper were modified by the catalyst). The bulk electrolysis was performed in an airtight electrochemical H-type cell with three electrodes. H-type cell consists of two compartments (volume of each part is 115 mL) separated by a Nafion®212 anion exchange membrane with 75 mL 0.5 M KHCO$_3$ electrolyte in each chamber and. Besides, LSV experiments were used with the catalyst-modified GCE as the working electrode. The preparation of the GCE working electrode is as follows. 6 mg electrocatalyst was added into 2 mL 0.5% Nafion solution. After sonication, 3 μL dispersed solution was dropped on GCE. The mass density of catalyst was 0.127 mg/cm$^2$. The electrochemical tests of different catalysts combinations were performed with full loading, i.e., the mass of the composite catalyst was 9 μg (Supplementary Table 4). The electrochemical surface area (ECSA) test of single catalyst components were performed using partial loading, i.e., 1% of 9 μg CDots and 6% of 9 μg Co$_3$O$_4$ (Supplementary Table 4). For LSVs experiments, initially, polarization curves for the modified electrode were carried out under an inert N$_2$ (gas) atmosphere. After this, the solution was purged with CO$_2$ (99.999%) for at least 30 min (CO$_2$-saturated high purity aqueous 0.5 M KHCO$_3$) and the electrocatalytic CO$_2$ reduction was measured.

**Data availability**. The data that support the findings of this study within the paper and its Supplementary Information file are available from the corresponding authors on request.

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

## Acknowledgments

This work was supported by the Collaborative Innovation Center of Suzhou Nano Science and Technology, the National Natural Science Foundation of China (51725204, 21771132, 51422207, 51572179, 21471106, and 21501126), the Natural Science Foundation of Jiangsu Province (BK20161216), and a project funded by the Priority Academic Program Development of Jiangsu Higher Education Institutions (PAPD).

## Author contributions

Z.K. designed and supervised the project. Y.L. partly designed and supervised the project. Y.L. and J.X. supervised parts of the project. S.G. conducted the synthesis. S.Z. performed the test of the electrochemical surface area. S.Z. and X.W. performed the BET measurements. S.G. carried out all remaining electrochemical measurements and the catalysts characterizations. S.G. wrote the manuscript. N.Y. contributed to the data analysis of the i–t curves in the $CO_2$-saturated $KHCO_3$ solution and the PB solution. Y.L. and Z.K. modified and finalized the manuscript. All authors contributed to data analysis and approved the final version of the manuscript.

## Additional information

**Competing interests:** The authors declare no competing financial interests.

