## [Peer Review File · Nature Communications]

Reviewers' comments:

Reviewer #1 (Remarks to the Author):

This work reports an interesting observation of tunable catalytic activity for H₂/CO generation using composite-based catalysts and the generation of the synthesis approach of composite. This work seems to be performed systematically and would be potentially useful in the related community, However, several arguments and explanation are not fully supported in the current manuscript, which requires additional data to clarify the issues.

The presence of CDots on the composite was proven by TEM analysis, but that doesn't guarantee its presence. Even if so, I wonder how homogeneously the CDots are distributed throughout the composite, which is essential for effective catalytic performance. According to Figure 2, it is critical for CDots and Co₃O₄ to be located in the near vicinity for the catalytic reaction, so I think the authors need to do more rigorous investigation on TEM analysis to clarify the presence of CDots and the distribution of Co₃O₄ and CDots. Also, unlike the small particles in the TEM image, the diffraction peaks from Co₃O₄ are quite sharp, which may imply the presence of much larger particles than seen in the TEM image.

The Co species in the XPS spectrum are wrongly assigned. The chemical shift in the cobalt compounds cannot be used for discriminating the valence state of Co because of its complexity and satellite peaks. Typically, Co²⁺ has a higher binding energy than Co³⁺ in many cobalt compounds, which is opposite to the current assignment.

The way to determine H₂ adsorption by HCl is not described in the manuscript.

H₂ and CO production on C₃N₄ are proportional to the applied bias, but this trend gets reverse on CDots-C₃N₄. Can they provide reasons for this observation?

CDots anchored on C₃N₄ looks to be a booster for CO generation, but once CO₂ is adsorbed on CDots, CO₂ needs to diffuse off to C₃N₄ for the reduction. So does the adsorbed H radical. I wonder whether they can provide some information about the relative strength of adsorption of these species. Also, is there any possibility for CDots on C₃N₄ to act as the catalytic center for CO₂ reduction? Because two reactants (CO₂ and H radical) are abundant on the CDots, there may be no need for the diffusion of these adsorbed species to C₃N₄.

Reviewer #2 (Remarks to the Author):

Reject. This manuscript introduced a concept of a ternary catalyst design for CO₂ reduction to syngas generation. A high stability is achieved for the process. However there are still several questions as suggested below:

1. There is not enough evidence in the Figure 1a. to confirm the mesoporous structure of the the Co₃O₄-CDots-C₃N₄ as suggested by the authors. More experimental data or other TEM images are needed.
2. A better understanding of the mechanism might be needed. According to Figure 4, CDots can produce hydrogen on the electrode as well. The improvement of the Co₃O₄ is not clear. Can syngas generated on CDots-C₃N₄?
3. One of the advantages of this concept is to get a catalyst design which has one part generate for H₂ only and the other part just produce CO. What is the advantage of this design? As showing by Sun et. al (J. Am. Chem. Soc. 2014, 136, 16132-16135.), a smaller overpotential and higher CO generation is achieved on single Au nanowire catalyst. The ratio of CO/H₂ is also tuneable.
4. Line 286, Figures 3a, 4c and 4d didn't show the information the authors trying to explanation in the context.

Reviewer #3 (Remarks to the Author):

In this manuscript, the authors reported an investigation on the synthesis of a hybrid material for stimulating CO₂ and water splitting to produce syngas with tunable ratio of Co/H₂. This study is situated in the frame of the relevant interest shown by the scientific community in looking for earth abundant, cheap and stable catalysts for the production of clean chemical fuels or raw materials from CO₂ and H₂O. The materials are well characterized and this manuscript is well organized. However, there are several literatures on the design, synthesis and applications of nanocomposites with carbon materials and oxides for stimulating CO₂ and water splitting. The authors need to state very clearly which part of results presented here are beyond those studies.

In addition, the authors need to clarify the following unclear points.

Did authors consider the synergistic effect between CO₂ reduction and HER? The actual role of each component is individually clarified, but the synergistic role is not fully clarified. Because several studies are currently on-going in different fields of applications, I think authors should better point out in this manuscript.

Experimental section is moreover not completed.

The authors need to correct some typos in the manuscript.

Detailed response to the reviewers' comments

Reviewer #1 (Remarks to the Author):

This work reports an interesting observation of tunable catalytic activity for H₂/CO generation using composite-based catalysts and the generation of the synthesis approach of composite. This work seems to be performed systematically and would be potentially useful in the related community. However, several arguments and explanation are not fully supported in the current manuscript, which requires additional data to clarify the issues.

Our response:

We thank the reviewer for his positive evaluation of our work and for his comments the addressing of which contributed to the improvement of our manuscript.

Reviewer #1:

1. The presence of CDots on the composite was proven by TEM analysis, but that doesn't guarantee its presence. Even if so, I wonder how homogeneously the CDots are distributed throughout the composite, which is essential for effective catalytic performance. According to Fig. 2, it is critical for CDots and Co₃O₄ to be located in the near vicinity for the catalytic reaction, so I think the authors need to do more rigorous investigation on TEM analysis to clarify the presence of CDots and the distribution of Co₃O₄ and CDots.

Our response:

The composition, structure and morphology of the Co₃O₄-CDots-C₃N₄ composites was determined by a variety of characterizations. EDX and XPS has shown it includes Co, O, C and N (EDX: ~1.6 at% Co, 3 at% O, 49 at% N and 47 at% C; XPS: ~1.8 at% Co, 5.8 at% O, 52.5 at% N and 39.9 at% C. The slight elemental differences are due to the errors introduced by the probed samples (e.g. use of In as a substrate for the XPS measurements). XRD (Supplementary Fig. 2) has shown that it contains Co₃O₄. Upon the reviewer's request we have added more TEM and HRTEM images to better validate the distribution of Co₃O₄ and CDots. TEM (Fig. R1a) shows nanoparticles evenly distributed in the matrix, which are mostly Co₃O₄ NPs judging from their contrast with respect to the matrix. Elemental mapping by EDS shows this matrix is made of C and N and reveals that Co is evenly distributed in this matrix (EDX cannot confirm the existence of CDots since the matrix consists C as well). HRTEM shows both graphitic carbon nanoparticles and Co₃O₄ nanoparticles. To further substantiate the existence of evenly

distributed CDots the reviewer is referred to the TEM imaging of a C_3N_4 -CDots composite (Supplementary Fig. 8c) in which no Co containing nanoparticles exist so that the CDots can be clearly identified. Similarly, CDots and metal containing nanoparticles can be clearly identified in each one of the other composites prepared (Supplementary Fig. 10).

Figure R1 (a) TEM image of Co_3O_4 -CDots- C_3N_4 composite; (b) HRTEM of Co_3O_4 -CDots- C_3N_4 composite.

Supplementary Figure 10 The TEM image and HRTEM image (inset) of (a) MoS_2 -CDots- C_3N_4 , (b) Pt-CDots- C_3N_4 , (c) Au-CDots- C_3N_4 , (d) h-Au-CDots- C_3N_4 . Typical CDots are circled by red ring.

Reviewer #1:

2. Also, unlike the small particles in the TEM image, the diffraction peaks from Co_3O_4 are quite sharp, which may imply the presence of much larger particles than seen in the TEM image.

Our response:

The average size of Co_3O_4 NPs derived from the XRD spectrum (Supplementary Fig. 2) of the Co_3O_4 -CDots- C_3N_4 composites using the Debye-Scherrer equation

$$D = \frac{k\lambda}{B\cos\theta}$$

where, $k=0.89$, $\lambda=0.154178$ nm, $B=\text{FWHM}$ rad, $D=\text{particle size}$

The size of Co_3O_4 NPs is ~11-14 nm (see Table S1). This is in accord with the 5-10 nm sized Co_3O_4 nanoparticles observed by HRTEM.

Supplementary Figure 2 Large-angle XRD patterns of CDots- C_3N_4 composites (the red trace), Co_3O_4 -CDots- C_3N_4 composites (the black trace).

Supplementary Table 1 Co_3O_4 particle size derived from the XRD pattern using the Debye-Scherrer equation

No.	2θ	FWHM	FWHM/rad	Particle size
#1	31.4	0.63°	0.0101	14.1
#2	37	0.68°	0.0119	14.3

#3	45	0.71°	0.0122	12.1
#4	59.6	0.73°	0.0127	12.4
#5	65.4	0.82°	0.0143	11.3

Average size of Co₃O₄ NPs = 12.8 nm.

Reviewer #1:

3. The Co species in the XPS spectrum are wrongly assigned. The chemical shift in the cobalt compounds cannot be used for discriminating the valence state of Co because of its complexity and satellite peaks. Typically, Co²⁺ has a higher binding energy than Co³⁺ in many cobalt compounds, which is opposite to the current assignment.

Our response:

Typically, it is very well known that the binding energy increases with the degree of ionization. For Fe₃O₄ for example the binding energy (BE) of Fe³⁺ is larger than that of Fe²⁺. For Co₃O₄ there are contradictory assignments. In some BE(Co²⁺) > BE(Co³⁺) and in some BE(Co²⁺) < BE(Co³⁺). Initially we accepted the common concept that the binding energy increases with the degree of ionization. Following the reviewer's comment, we reconsidered the assignment and also consulted XPS experts that told us Co is a special case due to its special electronic configuration. We thus changed the peak assignment as suggested by the reviewer and thank him for his comment. The Co 2p XPS peak (formerly Fig. 1c) was transferred to Supplementary Fig. 4.

Reviewer #1:

4. The way to determine H⁺ adsorption by HCl is not described in the manuscript.

Our response:

The way to determine H⁺ adsorption by HCl was described in the supporting information Page 5. Following the reviewer's comment this part was revised and more clearly described.

The proton (H⁺) adsorption capacity of CDots, Co₃O₄, C₃N₄, CDots-C₃N₄, Co₃O₄-C₃N₄ and CDots-Co₃O₄-C₃N₄, was studied using the dialysis method applying a 5 mM HCl solution. A certain weight of catalyst (0.001 g CDots, 0.006 g Co₃O₄, 0.093 g C₃N₄, 0.5 g CDots-C₃N₄, 0.5 g Co₃O₄-C₃N₄ and 0.5 g CDots-Co₃O₄-C₃N₄) was added to a 50 mL 5 mM HCl solution. The CDots, Co₃O₄, C₃N₄, CDots-C₃N₄,

Co₃O₄-C₃N₄ or CDots-Co₃O₄-C₃N₄ solution was dialyzed using a semi-permeable membrane (MWCO 1000) in a 600 mL beaker and a 5 mM HCl (500 mL) dialysate. The adsorption of H⁺ by the catalyst decreases the amount of H⁺ in the catalyst compartment leading to a gradual crossing of H⁺ through the semi-permeable membrane and dialyze into the electrocatalyst solution. The dialysate was stirred for predetermined time intervals and then 2 mL dialysate was taken out for analysis and the concentration of the dialysate HCl solution was determined by titrating with a 5 mM NaOH solution. The single component catalysts weights were chosen to reflect the weight concentration of these catalysts in the Co₃O₄-CDots-C₃N₄ catalyst (1wt%CDots; 6wt%Co; 93wt%C₃N₄).

For single component (CDots, Co₃O₄ and C₃N₄), the amount of adsorbed H⁺ (Q_{single-component}, mg) by the above specified weight of the single component was calculated using the following equation:

$$Q = \frac{(C_0 - C_e) \times V}{1000/36.45}$$

where C₀ and C_e are the initial and temporal HCl concentrations (mg L⁻¹) respectively, V is the volume of HCl solution (500-2n mL, n is the number of temporal measurements), 36.45 is the molecular weight of HCl.

For the composites (CDots-C₃N₄, Co₃O₄-C₃N₄ and CDots-Co₃O₄-C₃N₄), the amount of adsorbed H⁺ (Q_{composite}, mg g⁻¹) was calculated using the following equation:

$$Q = \frac{(C_0 - C_e) \times V}{1000W/36.45}$$

where C₀ and C_e are the initial and temporal HCl concentrations (mg L⁻¹) respectively, V is the volume of HCl solution (500-2n mL, n is the number of temporal measurements) and W is the weight (g) of composites. Note that Q_{single component} represents the total weight of H⁺ adsorbed per the fraction of single component catalyst in a CDots-Co₃O₄-C₃N₄ composite weighing 0.1 g. The expected adsorption of a mixture of these components (Q_{SC}(CDs) + Q_{SC}(Co₃O₄) + Q_{SC}(C₃N₄) is compared to that of 0.1g of the CDots-Co₃O₄-C₃N₄ composite (0.1 g Q_{composite}). The actual adsorption of 0.1g composite was 1.38 mg, almost twice larger than the expected adsorption of the pure components which was 0.787 mg. This indicates that the chemical blending of the pure components to a composite catalyst has a positive synergistic effect on the H⁺ adsorption.

Reviewer #1:

5. H₂ and CO production on C₃N₄ are proportional to the applied bias, but this trend gets reverse on CDots-C₃N₄. Can they provide reasons for this observation?

Our response

The reviewer is referring to Fig. 4b in which the Faradaic efficiency (FE) of H₂ and CO production on C₃N₄ slightly increases with the applied bias while FE of CO production on CDots-C₃N₄ indeed decreases with the bias (FE of H₂ production on CDots-C₃N₄ increases with bias and does not decrease). Note however that the actual production rate is described by the current density which has two parts, j_{H₂} and j_{CO} and not by the FEs. Fig. 3a and 3b clearly show that both current densities increase with bias while their corresponding FEs may either increase with or decrease with bias depending on the relative efficiency of a specific channel (CO or H₂ generation). Also note that the generation rate of H₂ and CO in both C₃N₄ (curve 1 in Fig. 4a) and CDots-C₃N₄ (curve 4 in Fig. 4a) are very small! They both increase with bias.

Reviewer #1:

6. CDots anchored on C₃N₄ looks to be a booster for CO generation, but once CO₂ is adsorbed on CDots, CO₂ needs to diffuse off to C₃N₄ for the reduction. So does the adsorbed H radical. I wonder whether they can provide some information about the relative strength of adsorption of these species. Also, is there any possibility for CDots on C₃N₄ to act as the catalytic center for CO₂ reduction? Because two reactants (CO₂ and H radical) are abundant on the CDots, there may be no need for the diffusion of these adsorbed species to C₃N₄.

Our response:

CDots per se do not act as a catalytic center for CO₂ reduction as evident from Fig. 4b. CO₂ is adsorbed by both CDs and its composite with C₃N₄ (CDots-C₃N₄ and Co₃O₄-CDots-C₃N₄). The relative amount of CDs in the composite is only about 1wt%, so that the total amount of CO₂ adsorbed on C₃N₄ is much larger than that adsorbed on CDs (Supplementary Fig. 5a). CO₂ is abundant on C₃N₄ and does not need to diffuse to C₃N₄. Similarly, H⁺ is abundant on C₃N₄ which is the major component of the composite catalyst. The role of the CDs is to attract H⁺ and electrons to form and stabilize H•. The H• radicals diffuse to CO₂ adsorbed on C₃N₄ in its vicinity (the CDs are dispersed on C₃N₄ so the diffusion length required is only a few nm) and C₃N₄ catalyzes the reduction of CO₂ (CO₂+2H•→CO+H₂O). The H• radicals may alternatively diffuse to Co₃O₄ where H₂ molecules are generated. The short distance between the CDs and the Co₃O₄ guarantees that the H• radicals indeed reach the appropriate H₂

generation site. Note that the incorporation of CDots in the porous C_3N_4 surfaces increases the adsorption of both CO_2 and H^+ on C_3N_4 in a synergistic way. This is why the adsorption on the 1wt%CDs-99wt% C_3N_4 composite is much larger than 1% adsorption of pure CDots + 99% adsorption of pure C_3N_4 (Supplementary Fig. 5 and 6).

Following the reviewer's question about the adsorption of CO_2 and H^+ on the different components of our catalyst we have measured (Supplementary Fig. 5 and 6) the adsorption on the individual components (CDots, Co_3O_4 , and C_3N_4) as well as on their different composite combinations (CDots- C_3N_4 , Co_3O_4 - C_3N_4 , CDots- Co_3O_4 and Co_3O_4 -CDots- C_3N_4). C_3N_4 is the main component (> 90 wt.%) of the composite so it has the largest contribution to the adsorption. The synergism of the three components acting simultaneously enhances the H^+ and CO_2 adsorption by a factor of 2-3 with respect to the adsorption of the individual components.

Supplementary **Figure 5** CO_2 adsorption on the different components of the catalysts. (a) CO_2 adsorption by 0.001 g CDots, 0.006 g Co_3O_4 and 0.093 g C_3N_4 ; (b) CO_2 adsorption ($mmol\ g_{catalyst}^{-1}$) of the composite catalyst. The total CO_2 adsorption of the components in Supplementary Fig. 5a ($CO_2(0.001g\ CDots) + CO_2(0.006\ g\ Co_3O_4) + CO_2(0.093\ C_3N_4) = 0.00345\ mmol + 0.00069\ mmol + 0.00848\ mmol$) is 0.01262 mmol. The adsorption of H^+ by 0.1g Co_3O_4 -CDots- C_3N_4 in Supplementary Fig. 5b is 0.0332 mmol. This indicates that the synergistic adsorption of the composite is larger (by a factor of 3) than the individual adsorption of each component.

Supplementary Figure 6 H⁺ adsorption on the different components of the catalysts. (a) H⁺ adsorption by 0.001 g CDots, 0.006 g Co₃O₄ and 0.093 g C₃N₄; (b) H⁺ adsorption (mg g_{catalyst}⁻¹) of the composite catalyst. The total H⁺ adsorption of the components in Supplementary Fig. 6a (H⁺(0.001g CDots) + H⁺(0.006 g Co₃O₄) + H⁺(0.093 C₃N₄) = 0.212 mg + 0.041 mg + 0.534 mg) is 0.787 mg. The adsorption of H⁺ by 0.1g Co₃O₄-CDots-C₃N₄ in Supplementary Fig. 6b is 1.38 mg. This indicates that the synergistic adsorption of the composite is larger (by a factor of 2) than the individual adsorption of each component.

Reviewer #2 (Remarks to the Author):

Reject. This manuscript introduced a concept of a ternary catalyst design for CO₂ reduction to syngas generation. A high stability is achieved for the process. However, there are still several questions as suggested below:

Our response:

We thank the reviewer for recognizing the novelty of our work and are happy to answer his questions.

Reviewer #2:

1. There is not enough evidence in the Fig. 1a. to confirm the mesoporous structure of the Co₃O₄-CDots-C₃N₄ as suggested by the authors. More experimental data or other TEM images are needed.

Our response:

The reviewer is correct. The TEM data of Fig. 1a shows the (rough) morphology and the composition of the Co_3O_4 -CDots- C_3N_4 composite, but not its porosity. The porosity indicators (the BET surface area and the pore size distribution of the catalyst (Fig. 1c)) were determined from the adsorption isotherm of N_2 (Fig. 1d) at liquid N_2 temperature (77 K) obtained using a Micromeritics 2020 instruments and their data processing algorithms. The BET surface area of the catalyst is $158.7 \text{ m}^2 \text{ g}^{-1}$ and its pore size distribution (Fig. 1c insert) is peaked around 35-50 nm. Note that the TEM data of Supplementary Fig. 9 and 11 also show a morphology of a porous material. The text was corrected accordingly.

Reviewer #2:

2. A better understanding of the mechanism might be needed. According to Fig. 4, CDots can produce hydrogen on the electrode as well. The improvement of the Co_3O_4 is not clear. Can syngas generate on CDots- C_3N_4 ?

Our response

CDots attract H^+ and electrons and stabilize $\text{H}\bullet$. The amount of $\text{H}\bullet$ would increase with bias so that some H_2 is formed. Fig. 4a curve 2 however indicates that the amount of H_2 generated by CDots is much smaller than that obtained by adding Co_3O_4 (curve 6). Co_3O_4 seems to be a poor HER catalyst once it is acting alone or with C_3N_4 (curves 3 and 5) and its H_2 production is boosted by the CDots which stabilize and provide $\text{H}\bullet$. Note that the low amounts of H_2 produced by pure CDots in curve 2 are generated using a CDots areal mass density ~ 100 times larger than the amount of CDots used in the Co_3O_4 -CDots- C_3N_4 composite. Similarly, the seemingly large amount of H_2 produced by CDots+ Co_3O_4 (curve 6) are generated using a CDots mass density ~ 100 times larger than the amount of CDots used in the Co_3O_4 -CDots- C_3N_4 composite (so that **per g CDots** the HER activity of the Co_3O_4 -CDots- C_3N_4 composite (curve 7) is much larger than that of curve 6). A very low amount (low current density of 1 mA cm^{-2} or less, see Fig. 4a curve 4) of syngas can be produced on CDots- C_3N_4 and the CO/H_2 ratio is 8:1 and 3.5:1 for -0.5 and -0.8 V respectively, i.e. the relative amount of H_2 is small.

Reviewer #2:

3. One of the advantages of this concept is to get a catalyst design which has one part generate for H_2 only and the other part just produce CO. What is the advantage of this design? As showing by Sun et. al (J.

Am. Chem. Soc. 2014, 136, 16132-16135.), a smaller overpotential and higher CO generation is achieved on single Au nanowire catalyst. The ratio of CO/H₂ is also tunable.

Our response:

We thank the reviewer for drawing our attention to the works of Sun et al (JACS 2013, 135, 16833-16836; JACS 2014, 136, 16132-16135). These works report the reduction of CO₂ to CO applying Au nanoparticles or nanowires. The works of Sun et al are focused on the generation of CO from CO₂ attempting to reach a selectivity exceeding 90% and approaching 100%. They very briefly mention the formation of H₂ as well but the topic of a stable tunable generation of syngas is not addressed at all. The origin of CO versus H₂ generation is associated with edge sites (CO generation) or corner sites (H₂ generation). These are very specific reaction channels which cannot be applied in a general way to different catalysts. As the reviewer correctly comments the Au NWs catalyst produces CO at small overpotential and relatively large CO generation rate. The generation rate of syngas with large amounts of H₂ in Sun's work is however small.

The present work, as the reviewer himself comments, introduces a new and general concept of a ternary catalyst design for CO₂ generation and syngas production. The separation between the CO₂ reduction and the HER offers many simple and easy tunable ways to control the catalyst properties (balancing between CO₂ reduction and H₂ evolution). Syngas production using Co₃O₄-CDots-C₃N₄ was found tunable over a large range of H₂/CO ratios achieving a current density of 10 A g_{catalyst}⁻¹, much larger than in the work of Sun et al. Additionally, a high stability was achieved (as the reviewer indicates), much larger than that reported by Sun et al. The current density was found stable for at least 100 h (Fig. 3d) compared to a current reduction of 50% within a few hours in the work of Sun et al. The FE was stable for at least 30 h (Fig. R6). It is also interesting to note that the application of Au to the ternary catalyst concept (i.e. h-Au-CDs-C₃N₄) significantly reduces the amount of Au needed (with respect to Sun's work) to only 0.6 wt% of the catalyst yielding a very high current density of 708 A g_{Au}⁻¹ for CO generation and 747 A g_(Au)⁻¹ for H₂ generation under conditions yielding CO/H₂~1 (i.e. together the A g_{Au}⁻¹~1450). Such a significant reduction of the amount of Au needed for the same amount of CO generated significantly reduces the cost of the electrodes.

Additional significant aspects of our work are: (1) the elucidation of the different mechanisms which govern the performance of the ternary catalysts; (2) the demonstration of the validity of the new design concept for a variety of different HER catalysts, (3) the possibility of adopting the ternary catalyst concept to other reactions.

Following the reviewer's comment, we refer to the works of Sun et al in the modified manuscript and point out the advantages of our system to his in terms of a much better stability and a higher mass activity of our ternary catalysts ($10 \text{ A g}_{\text{catalyst}}^{-1}$ for $\text{Co}_3\text{O}_4\text{-CDots-C}_3\text{N}_4$ and $>700 \text{ A g}_{\text{Au}}^{-1}$ for both CO and H_2 generation for h-Au-CDots- C_3N_4).

Figure R6 The stability of the current density and the CO Faradaic efficiency for 30h of continuous operation.

Reviewer #2:

4. Line 286, Fig. 3a, 4c and 4d didn't show the information the authors trying to explanation in the context.

Our response:

The reviewer is correct. We had a typo in line 286. It was "Fig. 3a, 4c and 4d show the j_{H_2} and j_{CO} obtained at different applied potentials for 0.05, 0.04, and 0.02 M". It was corrected to: "Fig. 3a, 4e and 4f show the j_{H_2} and j_{CO} obtained at different applied potentials for 0.05, 0.04, and 0.02 M $\text{Co}_3\text{O}_4\text{-CDots-C}_3\text{N}_4$ composites, respectively".

Reviewer #3 (Remarks to the Author):

1. In this manuscript, the authors reported an investigation on the synthesis of a hybrid material for stimulating CO_2 and water splitting to produce syngas with tunable ratio of CO/H_2 . This study is situated in the frame of the relevant interest shown by the scientific community in looking for earth abundant, cheap and stable catalysts for the production of clean chemical fuels or raw materials from CO_2 and H_2O . The materials are well characterized and this manuscript is well organized.

Our response:

We do thank reviewer's positive and valuable comments and we have modified the manuscript accordingly.

Reviewer #3:

2. However, there are several literatures on the design, synthesis and applications of nanocomposites with carbon materials and oxides for stimulating CO₂ and water splitting. The authors need to state very clearly which part of results presented here are beyond those studies.

Our response:

The reviewer is correct that nanocomposites of carbon based and oxides were used for either CO₂ reduction or HER applications. Our current work, however, presents a novel concept of a ternary electrocatalyst system in which CDs act as a balancing valve between two different catalysts activating two different reaction channels: (1) a HER catalyst and (2) a CO₂ reduction catalyst. To the best of our knowledge such a system is presented for the first time. The particular application that we demonstrate is tunable, stable syngas production. We have shown that the concept is a general one and might be complemented applying different HER catalysts. The same concept might be used to balance between other reaction channels (i.e. other important gas compositions relevant to the chemical industry and not just syngas). Note that the synergistic effects of a composite material while being useful, may make the system complex and difficult to control. Our concept is to design a system in which each of the two balanced catalysts works independently (as a zero order approximation) and the synergistic effects are minimal so that the system becomes stable and easy to control. Following the reviewer's request, we noted in the revised manuscript that CO₂ reduction to CO and hydrogen evolution reactions (HER) per se are two independent major and important fields and explained that we introduce a new concept for syngas generation.

Reviewer #3:

3. In addition, the authors need to clarify the following unclear points. Did authors consider the synergistic effect between CO₂ reduction and HER? The actual role of each component is individually clarified, but the synergistic role is not fully clarified. Because several studies are currently on-going in different fields of applications, I think authors should better point out in this manuscript.

Our response:

The present manuscript presents the concept of a ternary composite catalyst in which as a zero approximation the CO₂ reduction and the HER channels are independent and are balanced by the CDots which act as a valve. The rationale of this design is that the synergism between HER and CO₂ reduction complicates the system and makes it difficult to control. The success of achieving a tunable, stable and controllable system in which the role of each component is well defined indeed justifies this design concept.

Our experiments nevertheless show that the ternary system does exhibit second order synergistic effects. One such an effect is the adsorption of both H⁺ and CO₂. The adsorption of the Co₃O₄-CDots-C₃N₄ is 2-3 times larger (Supplementary Fig. 5 and 6) than that of the sum of the adsorption of the individual catalyst components (Co₃O₄, CDots and C₃N₄). This indicates that the incorporation of CDots and Co₃O₄ nanoparticles into the C₃N₄ matrix does affect the H⁺ and CO₂ adsorption in a synergistic way, which in turn affects the H₂ and CO₂ generation.

In the ternary catalyst the HER and CO₂ reduction compete with each other over the same H• species which are essential for the two reactions. The increase of one reaction thus causes the decrease of the other. A beneficial synergism between the two reactions increasing the total current of both is thus not expected, unless this synergism increases the total number of available H•. A more likely synergism might occur not between HER and CO₂ reduction, but between the three different catalyst components. One example discussed above is the increased adsorption of both H⁺ and CO₂. Another example is the extremely high mass activity of Au obtained applying the h-Au-CDots-C₃N₄ system of >700 A g_{Au}⁻¹ which is two orders of magnitude larger than that of Au electrodes used for CO₂ reduction to CO (so that very small amounts of Au are needed for effective syngas generation by the Au-CDots-C₃N₄ system). A plausible explanation might be the proximity of the CDs (H• generation site) and C₃N₄ surfaces (CO₂ adsorption site and CO generation site) which decreases the losses of the diffusing species enhancing the mass activity.

Reviewer #3:

4. Experimental section is moreover not completed.

Our response:

We added additional information to the experimental section and revised the description of the BET measurements as well as the adsorption of H⁺ and CO₂ to the different catalysts to clarify these parts.

Reviewer #3:

5. The authors need to correct some typos in the manuscript.

Our response:

We have carefully read the manuscript and corrected all typos.

Reviewer #1 (Remarks to the Author):

I carefully read the responses to all the concerns raised by the reviewers and concluded that they make improvement to clarify most of the issues. I think it can be publishable.

Reviewer #2 (Remarks to the Author):

Suggestion: reject

The revised paper addresses some of the reviewer's concerns. The design of the electrode is novel, however, there is still lack of interpretation of experimental results and conclusions about mechanism. Specific comments:

-The interpretation given for Figure 4a is not clear. The results show that curve 6 has higher current density but curve 7 is suggested to be the most appropriate when considering catalyst loading. The surface areas should be based on the real surface area not the geometric surface area.

-Considering the complexity of the system, turnover frequencies and turn over numbers would give a direct comparison for the different catalysts reported here.

-Based on Figure S6, C-dots/C₃N₄ have higher adsorption toward CO₂ than H⁺ compared to C-dots/Co₃O₄/C₃N₄. A smaller onset potential is also found on C-dot/C₃N₄ compared to C-dots/Co₃O₄/C₃N₄ based on FigureS9. Is there a reason for why C-dots/Co₃O₄/C₃N₄ have the best behavior? Discussion here would be relevant.

Reviewer #3 (Remarks to the Author):

The authors have well addressed all comments/suggestions I raised. I recommend accepting the manuscript.

Detailed response to reviewer #2 comments

Reviewer # 2 general comments:

The revised paper addresses some of the reviewer's concerns. The design of the electrode is novel, however, there is still lack of interpretation of experimental results and conclusions about mechanism. Specific comments:

Our response: We thank the reviewer for recognizing the novelty of our design concept. In his previous evaluation he also acknowledged the high stability we achieve. We are sorry he feels that there is still lack of interpretation of experimental results and conclusions about the mechanism. Without reviewer #2 being specific we cannot further refer to this general comment. Note that we responded in details to all the concerns of reviewer #2 regarding the interpretations and conclusions raised in his previous evaluation. Also note that we have thoroughly explained both the experimental results and the mechanisms in the manuscript. The mechanisms were explained in two dedicated sections of the manuscript entitled: **“The three component electrocatalyst design and the proposed reaction mechanism for syngas generation”** and **“Investigation of the catalytic mechanisms”** and by a special figure (Figure 2) captioned: **“Schematic diagram of the reaction mechanisms”**.

Reviewer #2 comment 1

-The interpretation given for Figure 4a is not clear. The results show that curve 6 has higher current density but curve 7 is suggested to be the most appropriate when considering catalyst loading. The surface areas should be based on the real surface area not the geometric surface area.

Our response: Curve 6 in Figure 4a (derived from Co_3O_4 -Dots) has indeed a slightly higher current density than curve 7 (derived from Co_3O_4 -CDots- C_3N_4). The reaction product of curve 6 is however pure H_2 while the reaction product of curve 7 is syngas (H_2+O_2) as evident from figure 4b. The present work aims at the production of syngas which means that **curve 7 (Co_3O_4 -CDots- C_3N_4) is appropriate for syngas production while curve 6 is not**. Also note that the amount of the CDots and Co_3O_4 in the catalyst yielding curve 6 (Co_3O_4 -Dots) is ~ 100 times larger than in the catalyst yielding curve 7 (Co_3O_4 -CDots- C_3N_4) so that **the current-per(CDots and Co_3O_4)-catalyst-mass of curve 7 is actually 100 times smaller than that of curve 6**. This is fully explained in the manuscript and in the previous detailed response to reviewer #2. In choosing the quantities used for evaluation of the reaction rate of the different catalysts we have followed the host of previously reported syngas/ H_2 generation papers (Table R1) which use current densities based on geometrical area and not real area. This choice is practical for two reasons: (1) it allows evaluation of the gas generation current of electrodes with different geometrical sizes, (2) since the areal density of different catalysts is kept constant for comparison of different catalysts (using the geometrical surface area) the use of geometrical area allows calculation of the current-per-catalyst-mass.

Table R1. Summary of electrocatalysts for producing syngas from CO_2 and H_2O .

No.	Electrocatalysts	Quantity used to determine activity	Calculation method of activity	Ref.
1	Co_3O_4 -CDots- C_3N_4	7.7 mA/cm ² at -0.8 V	Using geometric surface area and the weight of catalyst	This work

2	10 nm C-Au NPs	~10 A/g at -0.7 V	Using the catalyst weight	¹
3	Ultrathin Au nanowires	~5.6 A/g at -0.5V	Using the catalyst weight and the geometric surface area	²
4	Ag/C ₃ N ₄	<-1 mA/cm ² at -0.6 V	Using the geometric surface area	³
5	Sr ₂ Fe _{1.5} Mo _{0.5} O _{6-δ}	-110 mA/cm ²	Using the geometric surface area	⁴
6	Re-functionalized GCE	-0.2 mA/cm ² at -1.5 V	Using the geometric surface area	⁵
7	Ru(II) polypyridyl carbene complex	-1.25 mA/cm ² at -1.4 V	Using the geometric surface area	⁶
8	Ag-based electrode	Not mentioned	Using the geometric surface area	⁷
9	Cu	-17 mA/cm ² at -1.2 V	Using the geometric surface area	⁸
10	Y ₂ O ₃ -doped ZrO ₂	-600 mA/cm ² at -1.25 V	Using the geometric surface area	⁹
11	Pd/C	-0.6 mA/cm ² at -0.7 V	Using the geometric surface area	¹⁰
12	M-N-C moieties	<25 mA/cm ² at -0.9 V	Using the geometric surface area	¹¹

References:

- Zhu W., et al. Monodisperse Au Nanoparticles for Selective Electrocatalytic Reduction of CO₂ to CO. *J. Am. Chem. Soc.* **135**, 16833-16836 (2013).
- Zhu W., et al. Active and Selective Conversion of CO₂ to CO on Ultrathin Au Nanowires. *J. Am. Chem. Soc.* **136**, 16132-16135 (2014).
- Sastre, F. *et al.* Efficient Electrochemical Production of Syngas from CO₂ and H₂O by using a Nanostructured Ag/g-C₃N₄ Catalyst. *ChemElectroChem* **3**, 1497–1502 (2016).
- Li, Y., Li, P., Hu, B. & Xia, C. A nanostructured ceramic fuel electrode for efficient CO₂/H₂O electrolysis without safe gas. *J Mater Chem A* **4**, 9236–9243 (2016).
- Zhou, X. *et al.* Graphene-Immobilized *fac* -Re(bipy)(CO)₃Cl for Syngas Generation from Carbon Dioxide. *ACS Appl. Mater. Interfaces* **8**, 4192–4198 (2016).
- Kang, P., Chen, Z., Nayak, A., Zhang, S. & Meyer, T. J. Single catalyst electrocatalytic reduction of CO₂ in water to H₂+CO syngas mixtures with water oxidation to O₂. *Energy Env. Sci* **7**, 4007–4012 (2014).
- Delacourt, C., Ridgway, P. L., Kerr, J. B. & Newman, J. Design of an Electrochemical Cell Making Syngas (CO+H₂) from CO₂ and H₂O Reduction at Room Temperature. *J. Electrochem. Soc.* **155**, B42 (2008).
- Kumar, B. *et al.* Controlling the Product Syngas H₂:CO Ratio through Pulsed-Bias Electrochemical

Reduction of CO₂ on Copper. *ACS Catal.* **6**, 4739–4745 (2016).

9. Chen, X., Guan, C., Xiao, G., Du, X. & Wang, J.-Q. Syngas production by high temperature steam/CO₂ coelectrolysis using solid oxide electrolysis cells. *Faraday Discuss* **182**, 341–351 (2015).

10. Sheng, W. *et al.* Electrochemical reduction of CO₂ to synthesis gas with controlled CO/H₂ ratios. *Energy Env. Sci* (2017). doi:10.1039/C7EE00071E

11. Varela, A. S. *et al.* Metal-Doped Nitrogenated Carbon as an Efficient Catalyst for Direct CO₂ Electroreduction to CO and Hydrocarbons. *Angew. Chem. Int. Ed.* **54**, 10758–10762 (2015).

Reviewer #2 comment 2

-Considering the complexity of the system, turnover frequencies and turn over numbers would give a direct comparison for the different catalysts reported here.

Our response: We follow a host of syngas/H₂ generation reported papers (see the list above) and compare the different catalysts by using a fixed catalyst areal density. We also use the same electrode size and geometry. Doing so the currents and current densities obtained for the different catalysts are comparable. We also calculate the current-per-catalyst-mass. These values allow also comparison to previously reported publications (e.g. the works of Sun *et al.*^{1,2} referred to us by reviewer #2). These values also assist in evaluation of the gas generation rates obtainable for practical applications and the production cost (molecules per \$ for specific catalysts). The use of TON or TOF is problematic since it involves the number of active sites which is difficult to assess for catalysts. Additionally, as indicated earlier, TON values are much less cited than currents/current densities/current per catalyst mass.

Reviewer #2 comment 3

-Based on Figure S6, C-dots/C₃N₄ have higher adsorption toward CO₂ than H⁺ compared to C-dots/Co₃O₄/C₃N₄. A smaller onset potential is also found on C-dot/C₃N₄ compared to C-dots/Co₃O₄/C₃N₄ based on FigureS9. Is there a reason for why C-dots/Co₃O₄/C₃N₄ have the best behavior? Discussion here would be relevant.

Our response: The CDots-C₃N₄ catalyst has a slightly higher adsorption towards CO₂ and H⁺ than Co₃O₄-CDots-C₃N₄ due to the negative effect of Co₃O₄ for adsorption (Figures S5 and S6). The current density (generating syngas) of the Co₃O₄-CDots-C₃N₄ system (Figure 4a and not S9) is however much larger than that of the CDots-C₃N₄ system (Figure 4a). This is due to the effect of the Co₃O₄ component (an effective hydrogen evolution reaction catalyst) which is missing in the CDots-C₃N₄ system. The Co₃O₄-CDots-C₃N₄ system allows separation between H₂ generation (on the HER catalyst serving as the generation site) and CO₂ reduction (on C₃N₄ serving as the CO generation site). This is the crucial point of our three component catalyst design which is thoroughly explained and discussed throughout the entire manuscript.

Reviewer #1 (Remarks to the Author):

This revised manuscript has been somewhat improved, but it still fails to fully address the concerns raised by the reviewer, which is quite important in evaluating the significance of this work. The specific catalytic activity should be compared on the basis of catalyst's surface area, not the geometric area of electrode. While many articles report the geometric area-based activity, it does not tell the accurate capability of catalyst. If the catalytic activity based on the actually surface area is lower, then one cannot claim this material is better. Also, the turnover frequencies and turn over numbers are indeed proper metrics for the comparison of catalytic activity. Ignoring this request by simply replying on the previous reports cannot be a good excuse for this request. As per the comment 3, the explanation is not good enough.

Reviewer #2 (Remarks to the Author):

The revised paper addresses some of the reviewer's concerns. The design of the electrode is novel, however, there is still lack of interpretation of experimental results and conclusions about mechanism. Specific comments:

-The interpretation given for Figure 4a is not clear. The results show that curve 6 has higher current density but curve 7 is suggested to be the most appropriate when considering catalyst loading. The surface areas should be based on the real surface area not the geometric surface area.

-Considering the complexity of the system, turnover frequencies and turn over numbers would give a direct comparison for the different catalysts reported here.

-Based on Figure S6, C-dots/C3N4 have higher adsorption toward CO₂ than H⁺ compared to C-dots/Co₃O₄/C3N₄. A smaller onset potential is also found on C-dot/C3N₄ compared to C-dots/Co₃O₄/C3N₄ based on FigureS9. Is there a reason for why C-dots/Co₃O₄/C3N₄ have the best behavior? Discussion here would be relevant.

Detailed response to the editor and to the reviewer comments

Editor comments:

Editor Comment 1:

While reviewer #1 previously stated that his/her own points were addressed, he/she thinks that, for a better comparison of catalytic activity, actual surface area, TON and TOF should be calculated and reported.

Our response to comment 1:

We adopted all recommendations of reviewer #1. We measured the real surface areas and calculated the real current densities and the TOFs (TONs are TOFs \times time). We did our best also to give a convincing discussion regarding why Co_3O_4 -CDots- C_3N_4 show a superior behavior (with respect to other combinations of CDots, Co_3O_4 and C_3N_4). It should be noted that as reviewer #2 commented we have a complex system. The use of quantities such as current density, mass activity, TOF and real surface areas in such a system consisting of three different catalysts with different functions is not trivial and requires special care.

Our Co_3O_4 -CDots- C_3N_4 catalyst is a composite of three different catalysts (Co_3O_4 , CDots, C_3N_4). We measured the LSVs of all the different combinations of these three catalysts (totally 7 combinations marked by curves 1-7 in Fig. 4a, modified to Fig. R1) in order to determine what is the role of each catalyst in the generation of syngas and what are the generation sites of H^\bullet , H_2 and CO. We have used for all these 7 catalyst combinations electrodes with the same geometrical area (3 mm in diameter electrode) and the same areal mass density of 0.127 g cm^{-2} . The results were presented and compared in terms of the current density (mA cm^{-2}) in which the geometrical area (0.0707 cm^2) was used to calculate the current density. To avoid the issue raised by the reviewers of what is the surface area that should be used (the real surface area or the geometrical surface area) we present the data in terms of the mass activity (current per gram catalyst) as shown in Figure R1.

Figure R1 The LSVs of the seven combinations of the different catalyst components. The y axis is expressed as: (i) current density (using geometrical surface area); (ii) mass activity (assuming that the

catalyst mass is the sum of the masses of all catalyst components). Note that the mass activity is independent of the surface area and reflects the real activity of each catalyst studied.

The calculation of mass activity allows to compare the activity of the different catalysts by a quantity which does not depend on the surface area of the catalyst. It is the approach most commonly used by researchers in the field and it also offers a way to evaluate the catalyst efficiency for practical applications (i.e. the reaction rate obtainable per unit mass of the catalyst). The derivation of the mass activity for curves 1-7 is however not trivial since we deal with three different catalyst components. The simplest calculation of the mass activity would be to consider the sum of the masses of the different catalysts and disregard their different nature and task (Fig. R1). Alternatively, we consider for each catalyst composition only the mass of one component (CDots, Co_3O_4 or C_3N_4) (Fig. R2 a-c). This enables comparison of the effect of the same amount of a specific component in the different catalyst combinations. For each component these figures show the evolution of the catalyst performance upon the addition of the other components. In all cases a two component system is more active than a one component system and the three components system is the most active.

Figure R2 The mass activity of the different combinations of the catalysts taking into account just one catalyst component indicated in the legend of the y-axis: **a** - Mass activity based on the weight of C_3N_4 in each combination of catalysts; **b** - Mass activity based on the weight of CDots in each combination of catalysts; **c** - Mass activity based on the weight of Co_3O_4 in each combination of catalysts. Note that the addition of components enhances the activity of the composite catalysts. The optimum catalyst composition in terms of the gas composition (H_2/CO) and current (syngas generation rate) is achieved for only $\text{Co}_3\text{O}_4\text{-CDots-}\text{C}_3\text{N}_4$.

The mass activity was further used to calculate the TOF which is related (and equivalent) to the mass activity. The TOF is the number of reacted electrons per unit time divided by the number of active catalytic sites. The number of active sites of the different catalysts is not known so that we approximated this number by the number of atoms of each (composite) catalyst. Since we have three different catalyst components with three different functions we calculate the TOF for only one component (Fig. 4a-4c) and compare the TOFs of all combinations of catalysts which contain this component (CDots, Co_3O_4 or C_3N_4). Similar to the analysis of the mass activity this presentation allows to evaluate the effect of the addition of the other components on the composite catalyst activity.

Figure R3 The TOFs of the different combinations of catalysts taking into account the number of atoms in just one component as representing the number of active sites of this component: **a** - TOFs based on the number of atoms of C_3N_4 in each combination of catalysts; **b** - TOFs based on the number of atoms of CDots in each combination of catalysts; **c** - TOFs based on the number of atoms of Co_3O_4 in each combination of catalysts. Note that the addition of components enhances the TOF of the composite catalysts. The optimum catalyst composition in terms of the gas composition (H_2/CO) and current (syngas generation rate) is achieved for only Co_3O_4 -CDots- C_3N_4 .

Following the reviewers request we measured the real surface area of all catalyst combinations using: (i) the BET method, (ii) an electrochemical method.

Table R1 Electrochemical surface area (ECSA) and BET surface area. S_r -real surface area, S_g -geometrical surface area. S_r/S_g = roughness factor. GCE - glassy carbon electrode. Full loading - the checked combination had a mass of $9 \mu g$ (loading 0.127 mg cm^{-2}). Partial loading - The checked single component had a mass of 1% of $9 \mu g$ (CDots), 6% of $9 \mu g$ (Co_3O_4) and (1% CDots + 6% Co_3O_4) μg representing the ECSA of a single component.

ECSA								
	GCE	C_3N_4	CDots- C_3N_4	Co_3O_4 - C_3N_4	Co_3O_4 -CDots- C_3N_4	CDots	Co_3O_4	CDots- Co_3O_4
Full loading		$9 \mu g$	$9 \mu g$	$9 \mu g$	$9 \mu g$	$9 \mu g$	$9 \mu g$	(1.3 CDots + 7.7 Co_3O_4) μg
Cdl (mF)	0.0072	0.040	0.043	0.042	0.045	0.0081	0.0128	0.0134
ECSA (cm^2)	0.18	1.82	1.95	1.91	2.05	0.37	0.32	0.34
S_r/S_g	2.55	25.7	27.6	27.0	29.0	5.23	4.53	4.81
Partial loading						$0.09 \mu g$	$0.54 \mu g$	(0.09 CDots + 0.54 Co_3O_4) μg
Cdl (mF)						0.0057	0.0066	0.0068
ECSA (cm^2)						0.032	0.04	0.045
S_r/S_g						0.45	0.57	0.64
BET surface area								
	GCE	C_3N_4	CDots- C_3N_4	Co_3O_4 - C_3N_4	Co_3O_4 -CDots- C_3N_4	CDots	Co_3O_4	CDots- Co_3O_4
S_r (m^2/g)		143.1	168.3	161.3	157.8	11.84	13.4	22.1

S_r (cm ²)		12.8	15.1	14.5	14.2	1.06	1.20	1.98
S_r/S_g		181	214	205	200	15.0	17.1	28.1

The BET derived values are larger by an order of magnitude than the electrochemical surface areas (ECSAs) in accord with previously reported data ($S_r/S_g \sim 200$ using BET compared to $S_r/S_g \sim 27$ using ECSA for the C_3N_4 containing catalysts; $S_r/S_g \sim 20$ using BET and $S_r/S_g \sim 5$ using ECSA for the nanoparticle catalysts). The measurements indicate that the four catalysts containing C_3N_4 have large and similar real surface areas while the real surface areas of the three nanoparticle catalysts (CDots, Co_3O_4 and CDots- Co_3O_4) are smaller by an order of magnitude. It is worth mentioning that the large real surface of C_3N_4 containing catalysts stabilizes the dispersion of CDots and Co_3O_4 nanoparticles and suppresses their agglomeration thus improving the composite catalyst activity. Additionally, these C_3N_4 surfaces along with the dispersed catalyst nanoparticles increase the adsorption of H^+ and CO_2 as was reported in the manuscript.

Note that to evaluate the ECSA one relies on unknown constants (e.g. C_s , the specific capacitance of the sample). Consequently, the precise determination of ECSA is difficult and the believable range of ECSA is within one order of magnitude (McCrorry, C. C. L., Jung, S., Peters, J. C. & Jaramillo, T. F. Benchmarking heterogeneous electrocatalysts for the oxygen evolution reaction, *J. Am. Chem. Soc.* **135**, 16977-16987 (2013)). One should also keep in mind that different methods of evaluating ECSA yield different results.

Determination of the real current density of the different catalyst compositions described in curves 1-7 should consider that they consist of three different catalysts with three different functions. Each component has its own, different, real surface area. We have thus measured the ECSA of each of the three components of the seven curves (CDots, Co_3O_4 , C_3N_4). We have then calculated the real current density per the ECSA of this component. The results are given in Fig. R4.

Figure R4 The real current density (current per catalyst component real area) of the different combinations of the catalysts taking into account the real area of just one catalyst component denoted in the y-axis: **a** - Real current density based on the ECSA of C_3N_4 in each combination of catalysts; **b** - Real current density based on the ECSA of CDots in each combination of catalysts; **c** - Real current density based on the ECSA of Co_3O_4 in each combination of catalysts. Note: Addition of catalyst components to an initial component increases the real current density which is optimized for the three component Co_3O_4 -CDots- C_3N_4 catalyst.

In contrast to the real current density, the mass activity (current per mass unit of the catalyst) analysis of different catalysts is independent of the surface area and, when each component is treated separately, yields a reliable comparative evaluation of the activity of the different catalysts compositions (Fig. R2). Similarly, the TOF analysis is independent of the surface area as well (Fig. R3). The figures of

the real current density (Fig. R4) have a large inaccuracy (one order of magnitude) but they still show, similar to the mass density and TOF curves, that the three component composite is the only optimal one for syngas generation.

Editor comment 2:

Also, it seems that more convincing discussion is needed in order to explain why CDots-Co₃O₄-C₃N₄ show a superior behavior.

Our response to comment 2:

Our goal in this work is the fabrication of a tunable, efficient catalyst for syngas (H₂+CO) generation. The new concept that we introduce is of a three component composite catalyst made of: (i) CDots which stabilize H• and deliver it to the two other components, (ii) C₃N₄, a catalyst which reduces CO₂ to CO (CO₂+2H•→CO+H₂O), (iii) Co₃O₄, a HER catalyst which generates H₂ (2H•→H₂). These three components are crucial to the efficient formation of syngas (Fig. R2 and R5). No CO is produced (Figure R5) in the absence of C₃N₄ (i.e. when using only CDots and Co₃O₄). The amount of H₂ produced without Co₃O₄ is negligible (when using CDots and C₃N₄). In the absence of CDots H• radicals are not delivered to the C₃N₄ or the Co₃O₄ and the amount of CO and H₂ generated is small (i.e. when using C₃N₄ and Co₃O₄). It is only the ternary system Co₃O₄-CDots-C₃N₄ which offers an optimized balance and efficiently (mass activity larger than 10A/g_{catalyst}) and generates syngas with H₂/CO ratios of 0.07:1 to 4:1.

Figure R2 The mass activity of the different combinations of the catalysts taking into account just one catalyst component indicated in the legend of the y-axis: **a** - Mass activity based on the weight of C₃N₄ in each composites; **b** - Mass activity based on the weight of CDots in each combination of catalysts; **c** - Mass activity based on the weight of Co₃O₄ in each combination of catalysts. Note that the addition of components enhances the activity of the composite catalysts. The optimum catalyst composition in terms of the gas composition (H₂/CO) and current (large syngas generation rate) is achieved for only Co₃O₄-CDots-C₃N₄.

Figure R5 The gas composition achieved by the different catalyst components applied. CO production necessitates C₃N₄. Significant H₂ production necessitates Co₃O₄.

Reviewer #1 comments and response to his comments see next page

Reviewers' comments:

Reviewer #1 (Remarks to the Author):

This revised manuscript has been somewhat improved, but it still fails to fully address the concerns raised by the reviewer, which is quite important in evaluating the significance of this work. The specific catalytic activity should be compared on the basis of catalyst's surface area, not the geometric area of electrode. While many articles report the geometric area-based activity, it does not tell the accurate capability of catalyst. If the catalytic activity based on the actually surface area is lower, then one cannot claim this material is better. Also, the turnover frequencies and turn over numbers are indeed proper metrics for the comparison of catalytic activity. Ignoring this request by simply replying on the previous reports cannot be a good excuse for this request. As per the comment 3, the explanation is not good enough.

Our response to reviewer #1:

We adopted all recommendations of reviewer #1. We measured the real surface areas of all the catalysts combinations using two independent methods: (i) BET, (ii) electrochemical surface area (ECSA) determination. We calculated the mass activity (current per the composite catalyst mass) of the different combinations of the catalysts (Fig. R1). We then considered the fact that each composite consists of different catalysts with different functions. We thus plotted three mass activity graphs (Fig. R2) in each of which the mass of only one specific component was considered (C_3N_4 , CDots or Co_3O_4). Similarly we calculated the TOFs of all catalysts combinations ($TON=TOF \times time$) taking into account each specific catalyst component (three graphs total, one for C_3N_4 , one for CDots and one for Co_3O_4). Note that the TOF is equivalent and related to the mass activity. TOF is the number of reacted electrons per unit time divided by the number of active sites. The number of reacted electrons per unit time is the current, while the number of active sites is unknown and is approximated by the number of catalyst atoms which is related to the catalyst mass.

Figure R1 The LSV of the seven combinations of the different catalyst components. The y axis is expressed as: (i) current density (geometrical surface area); (ii) mass density (assuming that the catalyst mass is the sum of the masses of all catalyst components). Note that the mass activity is independent of the surface area and reflects the real activity of each catalyst studied.

Figure R2 The mass activity of the different combinations of the catalysts taking into account just one catalyst component indicated in the legend of the y-axis: **a** - Mass activity based on the weight of C_3N_4 in each combination of catalysts; **b** - Mass activity based on the weight of CDots in each combination of catalysts; **c** - Mass activity based on the weight of Co_3O_4 in each combination of catalysts. Note that the addition of components enhances the activity of the composite catalysts. The optimum catalyst composition in terms of the gas composition (H_2/CO) and current (syngas generation rate) is achieved for only Co_3O_4 -CDots- C_3N_4 .

Figure R3 The TOF of the different combinations of catalysts taking into account the number of atoms in just one component as representing the number of active sites of this component. **a** - TOF based on the number of atoms of C_3N_4 in each combination of catalysts; **b** - TOF based on the number of atoms of CDots in each combination of catalysts; **c** - TOF based on the number of atoms of Co_3O_4 in each combination of catalysts. Note that the addition of components enhances the activity of the composite catalysts. The optimum catalyst composition in terms of the gas composition (H_2/CO) and current (syngas generation rate) is achieved for only Co_3O_4 -CDots- C_3N_4 .

Note that the calculation of mass activity allows to compare the activity of the different catalysts by a quantity which does not depend on the surface area of the catalyst. It is the approach most commonly used by researchers in the field which also offers a way to evaluate the catalyst efficiency for practical applications (i.e. the reaction rate obtainable per unit mass of the catalyst). The consideration of the mass of a single component in each combination of catalysts allows to observe that a single component is not active for syngas production, the addition of a second component enhances the activity but it is only the three component catalyst which is the optimal one.

Following the reviewers request we measured the real surface area of all catalyst combinations using: (i) the BET method, (ii) an electrochemical method (see Table R1).

Table R1 Electrochemical surface area (ECSA) and BET surface area. S_r -real surface area, S_g -geometrical surface area. S_r/S_g = roughness factor. GCE - glassy carbon electrode. Full loading – the checked combination had a mass of 9 μg (loading 0.127 mg cm^{-2}). Partial loading – The checked

single component had a mass of 1% of 9 μg (CDots), 6% of 9 μg (Co_3O_4) and (1% CDots + 6% Co_3O_4) μg representing the ECSA of a single component.

ECSA								
	GCE	C_3N_4	CDots- C_3N_4	Co_3O_4 - C_3N_4	Co_3O_4 - CDots- C_3N_4	CDots	Co_3O_4	CDots- Co_3O_4
Full loading		9 μg	9 μg	9 μg	9 μg	9 μg	9 μg	(1.3 CDots + 7.7 Co_3O_4) μg
Cdl (mF)	0.0072	0.040	0.043	0.042	0.045	0.0081	0.0128	0.0134
ECSA (cm^2)	0.18	1.82	1.95	1.91	2.05	0.37	0.32	0.34
S_r/S_g	2.55	25.7	27.6	27.0	29.0	5.23	4.53	4.81
Partial loading						0.09 μg	0.54 μg	(0.09 CDots + 0.54 Co_3O_4) μg
Cdl (mF)						0.0057	0.0066	0.0068
ECSA (cm^2)						0.032	0.04	0.045
S_r/S_g						0.45	0.57	0.64
BET surface area								
	GCE	C_3N_4	CDots- C_3N_4	Co_3O_4 - C_3N_4	Co_3O_4 - CDots- C_3N_4	CDots	Co_3O_4	CDots- Co_3O_4
S_r (m^2/g)		143.1	168.3	161.3	157.8	11.84	13.4	22.1
S_r (cm^2)		12.8	15.1	14.5	14.2	1.06	1.20	1.98
S_r/S_g		181	214	205	200	15.0	17.1	28.1

The BET derived values are larger by an order of magnitude than the electrochemical surface areas (ECSAs) in accord with reported data ($S_r/S_g \sim 200$ using BET compared to $S_r/S_g \sim 27$ using ECSA for the C_3N_4 containing catalysts; $S_r/S_g \sim 20$ using BET and $S_r/S_g \sim 5$ using ECSA for the nanoparticle catalysts). The measurements indicate that the four catalysts containing C_3N_4 have large and similar real surface areas while the real surface areas of the three nanoparticle catalysts (CDots, Co_3O_4 and CDots- Co_3O_4) are smaller by an order of magnitude. Note that to evaluate the ECSA one relies on unknown constants (e.g. C_s , the specific capacitance of the sample). Consequently, the precise determination of ECSA is difficult and the believable range of ECSA is within one order of magnitude (McCrory, C. C. L., Jung, S., Peters, J. C. & Jaramillo, T. F. Benchmarking heterogeneous electrocatalysts for the oxygen evolution reaction, *J. Am. Chem. Soc.* **135**, 16977-16987 (2013)). One should also keep in mind that different methods of evaluating ECSA yield different results.

In contrast, the mass activity (current per mass unit of the catalyst) analysis of different catalysts is independent of the surface area and, when each component is treated separately, yields a reliable comparative evaluation of the activity of the different catalysts compositions (Fig. R2). Similarly, the TOF analysis is independent of the surface area as well (Fig. R3). Finally, it is worth mentioning that the large real surface of C_3N_4 containing catalysts stabilizes the dispersion of CDots and Co_3O_4 nanoparticles and suppresses their agglomeration thus improving the composite catalyst activity. Additionally, these C_3N_4 surfaces along with the dispersed catalyst nanoparticles increase the adsorption of H^+ and CO_2 as was reported in the manuscript.

Determination of the real current density of the different catalyst compositions described in curves 1-7 should consider that they consist of three different catalysts with three different functions. Each component has

its own, different, real surface area. We have thus measured the ECSA of each of the three components of the seven curves (CDots, Co_3O_4 , C_3N_4). We have then calculated the real current density per the ECSA of this component. The results are given in Fig. R4.

Figure R4 The real current density (current per catalyst component real area) of the different combinations of the catalysts taking into account the real area of just one catalyst component denoted in the y-axis: **a** - Real current density based on the ECSA of C_3N_4 in each combination of catalysts; **b** - Real current density based on the ECSA of CDots in each combination of catalysts; **c** - Real current density based on the ECSA of Co_3O_4 in each combination of catalysts). Note: Addition of catalyst components to an initial component increases the real current density which is optimized for the three component Co_3O_4 -CDots- C_3N_4 catalyst only.

Reviewer #1 last comment

As per the comment 3, the explanation is not good enough.

Comment 3 (of reviewer #2) was:

Based on Figure S6, CDots/ C_3N_4 have higher adsorption toward CO_2 than H^+ compared to CDots/ Co_3O_4 / C_3N_4 . A smaller onset potential is also found on CDots- C_3N_4 compared to Co_3O_4 -CDots- C_3N_4 based on Figure S9. Is there a reason for why Co_3O_4 -CDots- C_3N_4 have the best behavior? Discussion here would be relevant.

Our response to last comment of reviewer #1 (comment 3 of reviewer #2)

CDots- C_3N_4 have indeed (Fig. R6) slightly higher adsorption toward CO_2 and H^+ compared to Co_3O_4 -CDots- C_3N_4 . This is due to the negative effect of Co_3O_4 on the adsorption, but the small reduction of adsorption has no significant effect on the activity of Co_3O_4 -CDots- C_3N_4 with respect to that of CDots- C_3N_4 .

Figure R6 (a) CO₂ adsorption. (b) H⁺ adsorption. CDots-C₃N₄ have a slightly lower CO₂ and H⁺ adsorption compared to Co₃O₄-CDots-C₃N₄ due to the negative effect of Co₃O₄.

The onset potential of CDots-C₃N₄ (-0.26 V) is however larger, not smaller (Fig. R7) than that of CDots-Co₃O₄-C₃N₄ (-0.24 V). Furthermore, the activity (generation rate) of syngas production of CDots-Co₃O₄-C₃N₄ is much higher (Figure R7) than that of CDots-C₃N₄. Moreover, while Co₃O₄-CDots-C₃N₄ produces syngas with H₂/CO ratio of 0.07:1 to 4:1, CDots-C₃N₄ produces syngas with low amounts of H₂ only (H₂/CO < 0.3:1). The reason is that CDots-C₃N₄ lacks an efficient HER catalyst (e.g. Co₃O₄) so that its hydrogen generation rate is small.

Our goal in this work is the fabrication of a tunable, efficient catalyst for syngas (H₂+CO) generation. The new concept that we introduce is of a three component composite catalyst made of: (i) CDots which stabilize H• and deliver it to the two other components, (ii) C₃N₄, a catalyst which reduces CO₂ to CO (CO₂+2H•→CO+H₂O), (iii) Co₃O₄, a HER catalyst which generates H₂ (2H•→H₂). These three components are crucial to the efficient formation of syngas. No CO is produced (Figure R5) in the absence of C₃N₄ (i.e. when using only CDots and Co₃O₄). The amount of H₂ produced without Co₃O₄ (Fig. R5) is negligible (when using CDots and C₃N₄). In the absence of CDots H• radicals are not delivered to the C₃N₄ or to Co₃O₄ and the amount of CO and H₂ generated is small (i.e. when using C₃N₄ and Co₃O₄). It is only the ternary system Co₃O₄-CDots-C₃N₄ which offers an optimized balance and efficiently (mass activity larger than 10 A/g_{catalyst}) generating syngas with H₂/CO ratios of 0.07:1 to 4:1. Fig. R2, R3 and R4 show the evolution of the mass activity, TOF and the real current density with the addition of each of the catalyst components and clearly demonstrate that it is only the full combination of Co₃O₄-CDots-C₃N₄, which provides the optimum performance for syngas production (i.e. large mass activity and syngas production with practical H₂/CO ratios).

Figure R5 The gas composition achieved by the different catalyst components applied. CO production needs C_3N_4 . Significant H_2 production needs Co_3O_4 .

Figure R7 The onset potential of $CDots-C_3N_4$ (-0.26 V) is larger, not smaller than that of $Co_3O_4-CDots-C_3N_4$ (-0.24 V). The current density of $Co_3O_4-CDots-C_3N_4$ (syngas generation rate) is much larger than that of $CDots-C_3N_4$ (H_2 generation rate, no CO).

Figure R2 The mass activity of the different combinations of the catalysts taking into account just one catalyst component indicated in the legend of the y-axis: **a** - Mass activity based on the weight of C_3N_4 in each combination of catalysts; **b** - Mass activity based on the weight of $CDots$ in each combination of catalysts; **c** - Mass activity based on the weight of Co_3O_4 in each combination of catalysts. Note that the addition of components enhances the activity of the composite catalysts. The optimum catalyst composition in terms of the gas composition (H_2/CO) and current (large syngas generation rate) is achieved for only $Co_3O_4-CDots-C_3N_4$.

Figure R3 The TOFs of the different combinations of catalysts taking into account the number of atoms in just one component as representing the number of active sites of this component. **a** - TOFs based on the number

of atom of C_3N_4 in each combination of catalysts; **b** - TOFs based on the number of atoms of CDots in each combination of catalysts; **c** - TOFs based on the number of atoms of Co_3O_4 in each combination of catalysts. Note that the addition of components enhances the activity of the composite catalysts. The optimum catalyst composition in terms of the gas composition (H_2/CO) and current (syngas generation rate) is achieved for only Co_3O_4 -CDots- C_3N_4 .

Figure R4 The real current density (current per real area of a single catalyst component) of the different combinations of the catalysts taking into account the real area of just one catalyst component denoted in the y-axis: **a** - Real current density based on the ECSA of C_3N_4 in each combination of catalysts; **b** - Real current density based on the ECSA of CDots in each combination of catalysts; **c** - Real current density based on the ECSA of Co_3O_4 in each combination of catalysts). Note: Addition of catalyst components to an initial component increases the real current density which is optimized for the three component Co_3O_4 -CDots- C_3N_4 catalyst only.

Reviewer #2 comments & response to his comments see next page !!!

Reviewer #2 (Remarks to the Author):

General comment

The revised paper addresses some of the reviewer's concerns. The design of the electrode is novel, however, there is still lack of interpretation of experimental results and conclusions about mechanism.

Our response to general comment reviewer #2:

We modified the manuscript and also added figures of the mass activity and TOF of the different combinations of catalysts as well as measured the real surface areas of these catalyst combination. We hope these additions assist in making the mechanisms and interpretation more clear.

Specific comments reviewer #2 :

Comment 1a reviewer #2

The interpretation given for Figure 4a is not clear. The results show that curve 6 has higher current density but curve 7 is suggested to be the most appropriate when considering catalyst loading.

Our response to comment 1a Reviewer #2:

Our goal in the work is to efficiently produce syngas with tunable ratios of H₂/CO. We achieve it by applying a three component composite catalyst Co₃O₄-CDots-C₃N₄ in which each component has a specific role: (i) CDots stabilize H• radicals and deliver them to the two other components, (ii) Co₃O₄ is the HER catalyst which generates H₂ (2H•→H₂), (iii) C₃N₄ reduces CO₂ to CO (CO₂+2H•→CO+H₂O). All these three components are crucial for efficient generation of syngas. Fig. R1 (modification of Fig. 4a) describes the activity of different compositions of the three components attempting to verify their role in the activity of the three component composite catalyst. Fig. R5 shows the gas composition achieved by the catalysts of Fig. R1.

Figure R1 The LSVs of the seven combinations of the different catalyst components. The y axis is expressed as: (i) current density (geometrical surface area); (ii) mass density (assuming that the catalyst mass is the sum of the masses of all catalyst components). Note that the mass activity is independent of the surface area and reflects the real activity of each catalyst studied.

Figure R5 The gas composition achieved by the different catalyst components applied. CO production needs C_3N_4 . Significant H_2 production needs Co_3O_4 .

Figure R2 The mass activity of the different combinations of the catalysts taking into account just one catalyst component indicated in the legend of the y-axis: **a** - Mass activity based on the weight of C_3N_4 in each combination of catalysts; **b** - Mass activity based on the weight of CDots in each combination of catalysts; **c** - Mass activity based on the weight of Co_3O_4 in each combination of catalysts. Note that the addition of components enhances the activity of the composite catalysts. The optimum catalyst composition in terms of the gas composition (H_2/CO) and current (syngas generation rate) is achieved for only Co_3O_4 -CDots- C_3N_4 .

Figure R3 The TOFs of the different combinations of catalysts taking into account the number of atoms in just one component as representing the number of active sites of this component: **a** - TOFs based on the number of atoms of C_3N_4 in each combination of catalysts; **b** - TOFs based on the number of atoms of CDots in each

combination of catalysts; **c** - TOFs based on the number of atoms of Co_3O_4 in each combination of catalysts. Note that the addition of components enhances the activity of the composite catalysts. The optimum catalyst composition in terms of the gas composition (H_2/CO) and current (syngas generation rate) is achieved for only $\text{Co}_3\text{O}_4\text{-CDots-C}_3\text{N}_4$.

Figure R4 The real current density (current per catalyst component real area) of the different combinations of the catalysts taking into account the real area of just one catalyst component denoted in the y-axis: **a** - Real current density based on the ECSA of C_3N_4 in each combination of catalysts; **b** - Real current density based on the ECSA of CDots in each combination of catalysts; **c** - Real current density based on the ECSA of Co_3O_4 in each combination of catalysts. Note: Addition of catalyst components to an initial component increases the real current density which is optimized for the three component $\text{Co}_3\text{O}_4\text{-CDots-C}_3\text{N}_4$ catalyst only.

The modified Fig. 4a (Fig. R1) represents the mass activity (current per gram catalyst) of all catalyst combinations. Curve 7 represents the activity per gram of $\text{Co}_3\text{O}_4\text{-CDots-C}_3\text{N}_4$, while curve 6 represents the activity per gram of $\text{Co}_3\text{O}_4\text{-CDots}$. The three catalyst components have different functions so a more correct description of their functional activity is to present: (1) the mass activity per specific catalyst component (Co_3O_4 , CDots or C_3N_4 see Fig. R2), (2) the TOF per specific catalyst component (Fig. R3) or (3) the real current density (current per real area of a specific catalyst component). In all cases it is obvious that the mass activity, TOF and real current density of curve 7 are much larger than those of curve 6. Additionally, it should be noted that **curve 7 represents production of syngas while curve 6 presents generation of H_2 only** (a HER catalyst is missing for $\text{Co}_3\text{O}_4\text{-CDots}$). So indeed, curve 7 is the most appropriate catalyst for syngas generation taking into account both the H_2/CO ratio (Fig. R5) and the mass activity (or TOF). Figs. R2-R4 that were added to the supplementary material clearly demonstrate that starting with a single catalyst component the addition of a second improves the catalytic properties but only the three component composite is the appropriate and optimized one for syngas production.

Reviewer #2 comment 1b:

The surface areas should be based on the real surface area not the geometric surface area.

Our response to Comment 1b Reviewer #2:

Following the reviewers request we measured the real surface area of all catalyst combinations using: (i) the BET method, (ii) an electrochemical method (see Table R1).

Table R1 Electrochemical surface area (ECSA) and BET surface area. S_r -real surface area, S_g -geometrical surface area. S_r/S_g = roughness factor. GCE - glassy carbon electrode. Full loading - the checked combination had a mass of 9 μg (loading 0.127 mg cm^{-2}). Partial loading - The checked single component had a mass

of 1% of 9 μg (CDots), 6% of 9 μg (Co_3O_4) and (1% CDots + 6% Co_3O_4) μg representing the ECSA of a single component.

ECSA								
	GCE	C_3N_4	CDots - C_3N_4	Co_3O_4 - C_3N_4	Co_3O_4 - CDots- C_3N_4	CDots	Co_3O_4	CDots- Co_3O_4
Full loading		9 μg	9 μg	9 μg	9 μg	9 μg	9 μg	(1.3 CDots + 7.7 Co_3O_4) μg
Cdl (mF)	0.0072	0.040	0.043	0.042	0.045	0.0081	0.0128	0.0134
ECSA (cm^2)	0.18	1.82	1.95	1.91	2.05	0.37	0.32	0.34
S_r/S_g	2.55	25.7	27.6	27.0	29.0	5.23	4.53	4.81
Partial loading						0.09 μg	0.54 μg	(0.09 CDots + 0.54 Co_3O_4) μg
Cdl (mF)						0.0057	0.0066	0.0068
ECSA (cm^2)						0.032	0.04	0.045
S_r/S_g						0.45	0.57	0.64
BET surface area								
	GCE	C_3N_4	CDots - C_3N_4	Co_3O_4 - C_3N_4	Co_3O_4 - CDots- C_3N_4	CDots	Co_3O_4	CDots- Co_3O_4
S_r (m^2/g)		143.1	168.3	161.3	157.8	11.84	13.4	22.1
S_r (cm^2)		12.8	15.1	14.5	14.2	1.06	1.20	1.98
S_r/S_g		181	214	205	200	15.0	17.1	28.1

The BET derived values are larger by an order of magnitude than the electrochemical surface areas (ECSAs) in accord with previously reported data ($S_r/S_g \sim 200$ using BET compared to $S_r/S_g \sim 27$ using ECSA for the C_3N_4 containing catalysts; $S_r/S_g \sim 20$ using BET and S_r/S_g using ECSA for the nanoparticle catalysts). The measurements indicate that the four catalysts containing C_3N_4 have large and similar real surface areas while the real surface areas of the three nanoparticle catalysts (CDots, Co_3O_4 and CDots- Co_3O_4) are smaller by an order of magnitude. Note that to evaluate the ECSA one relies on unknown constants (e.g. C_s , the specific capacitance of the sample). Consequently, the precise determination of ECSA is difficult and the believable range of ECSA is within one order of magnitude. (McCrorry, C. C. L., Jung, S., Peters, J. C. & Jaramillo, T. F. Benchmarking heterogeneous electrocatalysts for the oxygen evolution reaction, *J. Am. Chem. Soc.* **135**, 16977-16987 (2013)). One should also keep in mind that different methods of evaluating ECSA yield different results. In contrast, the mass activity (current per mass unit of the catalyst) analysis of different catalysts is independent of the surface area and, when each component is treated separately, yields a reliable comparative evaluation of the activity of the different catalysts compositions (Fig. R2). Similarly, the TOF analysis is independent of the surface area as well (Fig. R3). Finally, it is worth mentioning that the large real surface of C_3N_4 containing catalysts stabilizes the dispersion of CDots and Co_3O_4 nanoparticles and suppresses their agglomeration thus improving the composite catalyst activity. Additionally, these C_3N_4 surfaces along with the dispersed catalyst nanoparticles increase the adsorption of H^+ and CO_2 as was reported in the manuscript.

Determination of the real current density of the different catalyst compositions described in curves 1-7 should consider that they consist of three different catalysts with three different functions. Each component has its own, different, real surface area. We have thus measured the ECSA of each of the three component of the seven curves (CDots, Co_3O_4 , C_3N_4). We have then calculated the real current density per the ECSA of this component. The results are given in Fig. R4.

Figure R4 The real current density (current per catalyst component real area) of the different combinations of the catalysts taking into account the real area of just one catalyst component denoted in the y-axis: **a** - Real current density based on the ECSA of C_3N_4 in each combination of catalysts; **b** - Real current density based on the ECSA of CDots in each combination of catalysts; **c** - Real current density based on the ECSA of Co_3O_4 in each combination of catalysts. Note: Addition of catalyst components to an initial component increases the real current density which is optimized for the three component Co_3O_4 -CDots- C_3N_4 catalyst.

Reviewer # 2 comment 2:

Considering the complexity of the system, turnover frequencies and turn over numbers would give a direct comparison for the different catalysts reported here.

Our response to comment 2 reviewer #2:

The TOFs of the 7 curves were calculated and plotted for each specific catalyst component. The graphs were added to the supplementary material (Fig. R3).

Reviewer # 2 comment 3:

Based on Figure S6, CDots- C_3N_4 have higher adsorption toward CO_2 than H^+ compared to Co_3O_4 -CDots- C_3N_4 . A smaller onset potential is also found on CDot- C_3N_4 compared to Co_3O_4 -CDots- C_3N_4 based on Figure S9. Is there a reason for why Co_3O_4 -CDots- C_3N_4 have the best behavior? Discussion here would be relevant.

Our response to reviewer #2 comment 3:

CDots- C_3N_4 have indeed (Fig. R6) slightly higher adsorption toward CO_2 and H^+ compared to Co_3O_4 -CDots- C_3N_4 . This is due to the negative effect of Co_3O_4 on the adsorption, but its effect on the activity of Co_3O_4 -CDots- C_3N_4 with respect to that of CDot- C_3N_4 is small.

Figure R6 (a) CO₂ adsorption. (b) H⁺ adsorption. CDots-C₃N₄ have a slightly lower CO₂ and H⁺ adsorption compared to Co₃O₄-CDots-C₃N₄ due to the negative effect of Co₃O₄.

The onset potential of CDots-C₃N₄ (-0.26 V) is however larger, not smaller (Fig. R7) than that of Co₃O₄-CDots-C₃N₄ (-0.24 V). Furthermore, the activity (generation rate) of syngas production of Co₃O₄-CDots-C₃N₄ is much higher (Fig. R7) than that of CDots-C₃N₄. Moreover, while Co₃O₄-CDots-C₃N₄ produces syngas with H₂/CO ratio of 0.07:1 to 4:1, CDots-C₃N₄ produces syngas with low amounts of H₂ only (H₂/CO < 0.3:1). The reason is that CDots-C₃N₄ lacks an efficient HER catalyst (e.g. Co₃O₄) so that its hydrogen generation rate is small.

Our goal in this work is the fabrication of a tunable, efficient catalyst for syngas (H₂+CO) generation. The new concept that we introduce is of a three component composite catalyst made of: (i) CDots which stabilize H• and deliver it to the two other components, (ii) C₃N₄, a catalyst which reduces CO₂ to CO (CO₂+2H•→CO+H₂O), (iii) Co₃O₄, a HER catalyst which generates H₂ (2H•→H₂). These three components are crucial to the efficient formation of syngas. No CO is produced (Fig. R5) in the absence of C₃N₄ (i.e. when using only CDots and Co₃O₄). The amount of H₂ produced without Co₃O₄ (Fig. R5) is negligible (when using CDots and C₃N₄). In the absence of CDots H• radicals are not delivered to the C₃N₄ or Co₃O₄ and the amount of CO and H₂ generated is small (i.e. when using C₃N₄ and Co₃O₄). It is only the ternary system Co₃O₄-CDots-C₃N₄ which offers an optimized balance and efficiently (mass activity larger than 10 A/g_{catalyst}) generates syngas with H₂/CO ratios of 0.07:1 to 4:1. Fig. R2, R3 and R4 showing the evolution of the mass activity, the TOF and the real current density respectively with the addition of each of the catalyst components clearly show that it is only the full combination of Co₃O₄-CDots-C₃N₄, which provides the optimum performance for syngas production.

Figure R5 The gas composition achieved by the different catalyst components applied. CO production needs C_3N_4 . Significant H_2 production needs Co_3O_4 .

Figure R7 The onset potential of $CDots-C_3N_4$ (-0.26 V) is larger, not smaller than that of $Co_3O_4-CDots-C_3N_4$ (-0.24 V). The current density of $Co_3O_4-CDots-C_3N_4$ (syngas production rate) is much higher than the current density of $CDots-C_3N_4$ (H_2 production rate, no CO).

Figure R2 The mass activity of the different combinations of the catalysts taking into account just one catalyst component indicated in the legend of the y-axis: **a** - Mass activity based on the weight of C_3N_4 in each combination of catalysts; **b** - Mass activity based on the weight of CDots or the weight of CDots in each composites; **c** - Mass activity based on the weight of Co_3O_4 in each combination of catalysts. Note that the addition of components enhances the activity of the composite catalysts. The optimum catalyst composition in terms of the gas composition (H_2/CO) and current (large syngas generation rate) is achieved for only $Co_3O_4-CDots-C_3N_4$.

Figure R3 The TOFs of the different combinations of catalysts taking into account the number of atoms in just one component as representing the number of active sites of this component: **a** - TOFs based on the number

of atom of C_3N_4 in each combination of catalysts; **b** - TOFs based on the number of atoms of CDots in each combination of catalysts; **c** - TOFs based on the number of atoms of Co_3O_4 in each combination of catalysts. Note that the addition of components enhances the activity of the composite catalysts. The optimum catalyst composition in terms of the gas composition (H_2/CO) and current (syngas generation rate) is achieved for only Co_3O_4 -CDots- C_3N_4 .

Figure R4 The real current density (current per catalyst component real area) of the different combinations of the catalysts taking into account the real area of just one catalyst component denoted in the y-axis: **a** - Real current density based on the ECSA of C_3N_4 in each combination of catalysts; **b** - Real current density based on the ECSA of CDots in each combination of catalysts; **c** - Real current density based on the ECSA of Co_3O_4 in each combination of catalysts. Note: Addition of catalyst components to an initial component increases the real current density which is optimized for the three component Co_3O_4 -CDots- C_3N_4 catalyst.

Reviewer #1 (Remarks to the Author):

Most of the raised issues are addressed and the revision is by and large satisfactory.

Reviewer #2 (Remarks to the Author):

Authors have fully addressed the points raised before. I recommend publish without any changes.